# Exploiting the Chiral Ligands of *Bis*(imidazolinyl)- and *Bis*(oxazolinyl)thiophenes—Synthesis and Application in Cu-Catalyzed Friedel–Crafts Asymmetric Alkylation

**DOI:** 10.3390/molecules26237408

**Published:** 2021-12-06

**Authors:** Mohammad Shahidul Islam, Abdullah Saleh Alammari, Assem Barakat, Saeed Alshahrani, Matti Haukka, Abdullah Mohammed Al-Majid

**Affiliations:** 1Department of Chemistry, College of Science, King Saud University, P.O. Box 2455, Riyadh 11451, Saudi Arabia or 436106737@student.ksu.edu.sa (A.S.A.); ambarakat@ksu.edu.sa (A.B.); or 436106738@student.ksu.edu.sa (S.A.); 2Department of Chemistry, University of Jyväskylä, P.O. Box 35, FI-40014 Jyväskylä, Finland; matti.o.haukka@jyu.fi (M.H.)

**Keywords:** *bis-*oxazoline, *bis-*imidazoline, thiophene, indoles, *β*-nitroolefins, asymmetric catalysis, Friedel−Crafts alkylation

## Abstract

Five new *C*_2_-symmetric chiral ligands of 2,5-*bis*(imidazolinyl)thiophene (**L1****–L3**) and 2,5-*bis*(oxazolinyl)thiophene (**L4** and **L5**) were synthesized from thiophene-2,5-dicarboxylic acid (**1**) with enantiopure amino alcohols (**4a**–**c**) in excellent optical purity and chemical yield. The utility of these new chiral ligands for Friedel–Crafts asymmetric alkylation was explored. Subsequently, the optimized tridentate ligand **L5** and Cu(OTf)_2_ catalyst (15 mol%) in toluene for 48 h promoted Friedel–Crafts asymmetric alkylation in moderate to good yields (up to 76%) and with good enantioselectivity (up to 81% *ee*). The *bis*(oxazolinyl)thiophene ligands were more potent than *bis*(imidazolinyl)thiophene analogues for the asymmetric induction of the Friedel–Crafts asymmetric alkylation.

## 1. Introduction

Metal-catalyzed asymmetric transformation has become one of the most desirable strategies in advanced synthetic chemistry to access a variety of enantiopure organic molecules [1,2,3,4,5,6,7,8]. The optically active system can be achieved by means of various methodologies, such as chiral ligands assisted organocatalysis [9,10,11] and enzyme-catalyzed asymmetric conversion [12,13,14]. In addition, more advanced and refined approaches have been introduced effectively, such as stereo-convergent [15,16,17] and stereo-divergent synthesis [18,19,20,21] in order to acquire innumerable chiral frameworks.

Chiral ligand–Lewis acid metal complex-catalyzed asymmetric Friedel–Crafts alkylation reactions play a pivotal role in synthetic organic chemistry for the construction of new C–C bonds [22,23,24,25,26]. During the past few years, several chiral bidendate ligands have been developed and used in the Lewis acid metal-catalyzed asymmetric Friedel–Crafts alkylation reaction of indole with various substrates, including α,*β*-unsaturated-R-ketoesters (R = alkyl, aryl) [27,28], R-hydroxy enones (R = alkyl, aryl) [29,30], alkylidene malonates [31,32,33], acyl phosphonates [34,35], acyl heterocyclic compounds [36,37,38], *N*-sulfonyl aldimines catalyzed by Schiff base complexes of Cu(II)-chiral amino alcohol [39], α-trifluoromethylated *β*-nitrostyrenes catalyzed by chiral BINOL metal phosphate [40], nitroolefins catalyzed by oxazoline-imidazoline-Zn(II) [41], *bis*(oxazolinyl)-Cu(II) [42] and 2,5-*bis*(oxazolinyl)thiophenes-Cu(II) complexes [43]. Very recently, Tanaka et al. have documented homochiral metal–organic framework-catalyzed enantioselective Friedel–Crafts alkylation of *N*,*N*-dialkylanilines with trans-*β*-nitrostyrene [44]. That being said, very few examples of chiral metal–box-*bis*(oxazoline)/*bis*(imidazoline) complex-catalyzed enantioselective Friedel–Crafts alkylation of indole with nitroolefins have been documented to date [41,45,46,47].

In recent years, the application of nitroolefins as electrophiles has also been gaining notable interest among pharmacists due to the activation functionality of the nitro groups, which facilitate easy conversion to other useful functional groups to achieve numerous eye-catching chemical entities [48,49]. Furthermore, optically active Friedel–Crafts-alkylated product of indole with nitroolefins can also serve as an antecedent for the preparation of various drug molecules such as physostigmine [50,51], which acts as a clinically active anticholinergic drug [52], Recently, some examples of nitroalkenes have also been reported as Michael acceptors in metal-catalyzed asymmetric reaction due to the presence of strong electron-withdrawing nitro-groups [48,53,54] e.g., rhodium-catalyzed additions of boronic acids to nitroalkenes [55], copper-catalyzed dialkylzinc additions to nitroalkenes [56,57], conjugated reductions of nitroalkenes [58] and the organo-catalyzed additions of 1,3-dicarbonyl compounds to nitroalkenes [59,60].

Moreover, to date, most of the research work has been done with the main family of chiral ligands predominantly belonging to di-phosphine, diamine, di-ol, etc., i.e., phosphorous-, nitrogen- and oxygen-containing substrate. Very little research has been done in the recent past on developing chiral ligands based on sulfur-containing compounds. Therefore, researchers are highly interested in developing new chiral ligands based on a sulfur-containing moiety due to their high coordination ability to the most of the transition metals [61]. The sulfur atom is also considered as a soft atom that can bind strongly to soft metals, in particular copper metal Cu(II). In addition, sulfur-containing ligands are poor π-acceptors and poor σ-donors as compared to phosphine ligands, resulting in strong metal–sulfur bond strength. However, sulfur-containing ligand precursors are easily available, having extra advantages such as easy storage due to their higher tolerance to air as compared to phosphine-containing ligands, which makes them highly stable [61].

Recently, chiral ligand–Lewis acid-catalyzed asymmetric induction of indole with prochiral *β*-nitroolefin has become one of the most significant and successful pathways for accessing highly functionalized optically pure building blocks. Our research group has reported a new catalytic system based on the Cu(II) metal/chiral thiophene-2,5-*bis*(*β*-amino alcohol) ligands for an asymmetric Henry reaction of nitromethane with aromatic aldehyde with excellent *ee* (up to 94.6%) and chemical yield (up to 99%) [62]. In continuation of our research program, therefore, the design and synthesis of novel chiral 2,5-*bis*(imidazolinyl)thiophene and 2,5-*bis*(oxazolinyl)thiophene box-type ligands and their applications in various asymmetric catalyses remains a remarkable and interesting research topic to organic chemists. However, chiral ligands based on 2,5-*bis*(imidazolinyl)thiophene and 2,5-*bis*(oxazolinyl)thiophene framework could also be advantageous for several asymmetric transformations other than Friedel–Crafts alkylation reactions, such as asymmetric Henry reactions [63,64], Diels–Alder reactions [65,66], enantioselective additions of diethylzinc to acyclic enones [67,68,69], asymmetric allylic substitutions [70,71] and asymmetric cyclopropanation [72,73] reactions, etc. Keeping in mind the wide range of chiral applications of 2,5-*bis*(imidazolinyl)thiophene and 2,5-*bis*(oxazolinyl)thiophene box-type ligands and the diverse functionality of nitroolefins, we have decided to focus on this particular research field.

In this research article, we report the synthesis of novel chiral ligands thiophene-2,5-2,5-*bis*(imidazolinyl)thiophene (**L1–L3**) and thiophene-2,5-*bis*(oxazolinyl)thiophene (**L4** and **L5**) and their applications in Lewis acid metal-catalyzed asymmetric Friedel–Crafts alkylations of indole with electron-deficient prochiral *β*-nitroolefins.

Figure 1 shows some of the previously reported potent ligand structures used for asymmetric Friedel–Crafts alkylation reactions of indole with *β*-nitrostyrenes [41,42,62,74,75,76,77,78].

## 2. Results and Discussion

### 2.1. Synthesis of chiral 2,5-bis(imidazolinyl)thiophene (**L1****–L3**) and 2,5-bis(oxazolinyl)thiophene (**L4** and **L5**)

Two set of *C_2_*-symmetric 2,5-*bis*(imidazolinyl)thiophene (**L1–L3)** and 2,5-*bis*(oxazolinyl)thiophene (**L4** and **L5**) ligands, based on thiophene framework, were synthesized from readily available and cheap thiophene-2,5-dicarboxlyic acid (**1**) and chiral amino alcohols (**3a**–**c**) using well-known procedures reported in the literature [79] in five steps, as shown in Figure 1. At the very outset, thiophene-2,5-dicarboxlyic acid (**1**) was treated with thionylchloride (SOCl_2_) in the presence of a catalytic amount of *N*,*N*-dimethylformamide (DMF 2-3 drops) under reflux for 24 h, leading to the formation of acid chloride (**2**) in quantitative yields (crude), which was then allowed to react with three different amino alcohols (**3a**–**c**) in the presence of excess triethylamine (TEA) in dichloromethane (CH_2_Cl_2_) to produce thiophene-2,5-dicarboxamide alcohol derivatives **(4a**–**c)** with overall excellent isolated yield (75–97%). Thiophene-2,5-dicarboxamide alcohol (**4a)** was then refluxed in thionylchloride (SOCl_2_) for 24 h to afford crude thiophene-2,5-dicarboxamide dichloride (**5a**), which served as an intermediate for the synthesis of our target ligands **L1–L3**, while thiophene-2,5-dicarboxamide alcohol (**4b**–**c**) was chosen as the precursor for the synthesis of ligands **L4** and **L5** (Figure 2).

Ligands (**L1–L3)** were synthesized using the intermediate thiophene-2,5-dicarboxamide dichloride **5a** (2.92 mmol) by the reaction of three different aromatic amines **6a**–**c** (2.5 mmol) (aniline **6a**, *p*-chloroaniline **6b**, *p*-toludine **6c**) in the presence of excessive triethylamine (12 eq.) to form a corresponding thiophene-2,5-dicarboxamide intermediate (**7a**–**c**), which underwent a ring closure reaction upon treatment with 15% aqueous sodium hydroxide (NaOH) solution to form crude thiophene-2,5-*bis*(imidazolinyl)thiophene ligands (**L1–L3**)**.** Then, the ligands were further purified by column chromatography by eluting with EtOAc/petroleum ether/Et_3_N (*v**:v**:v* = 75:24:1) to afford pure ligands **L1–L3** (Figure 1). The isolated yields of the ligands were found to be in the range of 35–40%.

Under inert condition, ligands (**L4 and L5),** were prepared from thiophene-2,5-dicarboxamide alcohol (**4b and 4c)** by ring closure reaction upon being treated with tosylchoride (1.25 eq.) and triethylamine (4.0 eq.) in the presence of a catalytic amount of DMAP (cat. 0.1eq.) in dichloromethane (CH_2_Cl_2_) after 48 h of stirring at room temperature. The ligands were then purified by column chromatography, using 95% CH_2_Cl_2_/CH_3_OH as an eluent to afford pure ligands **L4** and **L5** (Figure 1) with 60% and 55% isolated yield, respectively. The formations of the compound thiophene-2,5-dicarboxamide alcohol (**3a)** and all the ligands (**L1****–L5**) were confirmed and characterized by NMR and mass spectroscopy analysis.

### 2.2. Application of Chiral Ligand **(L1****–L5**)

#### 2.2.1. Catalytic asymmetric Friedel–Crafts Alkylation of Indoles with Trans-*β*-nitrostyrene Derivatives; Optimization of Various Reaction Parameters

As soon as we had in our hand optically pure ligands **L1****–L5**, we decided to carry out the catalytic activity in an asymmetric Friedel–Crafts alkylation reaction between indoles **8a**–**d** and nitrostyrene derivatives **9a**–**h**. Indole (**8a**) and *p*-fluoronitrostyrene (**9a**) have been chosen as a model substrate for the reaction parameters optimization. In order to identify the best ligands for the asymmetric catalysis, initially, the Friedel–Crafts alkylation reaction of indole (**8a**) and *p*-fluoronitrostyrene (**9a**) was performed with the screened chiral *bis*(imidazoline) and *bis*(oxazoline) ligands **L1****–L5 (**15 mol%) and Cu(OTf)_2_ (15 mol%) as metal sources in toluene at room temperature for 48 h, and the subsequent findings are documented in Table 1. It is evident from the results summarized in Table 1, entries 1–5, that the thiophene-2,5-*bis*(oxazoline) ligand **L5** performed very well under the above-mentioned reaction conditions and afforded Friedel–Crafts alkylation adduct **10a** at 66% chemical yield with 75% enantiomeric excess (*ee*) (Table 1; entry 5), while ligand **L4** yielded 70% chemical yield with 45% *ee* (Table 1; entry 4). Although the ligands **L1–L3** furnished better chemical yields (78, 75 and 70%, respectively), only trace enantiomeric excess (*ee*) (3–5%) was achieved (Table 1, entries 1-3). In order to improve the chemical yield, the reaction was repeated with ligand **L5**, and reaction time was extended up to 72 h, but no significant changes were observed (Table 1; entry 6). Aiming to improve the chemical yield as well as enantioselectivity output of the reaction, a set of trials was conducted by variation of the loading of catalyst **L5**:Cu(OTf)_2_ at 5, 10 and 20 mol%. The results showed that regardless of the % catalyst loading, the chemical yield was lower (20%, 46% and 65%, respectively) and did not result in any significant changes for the enantioselectivity (65%, 71% and 74% *ee*) (Table 1, entries 7–9). The influences of the solvent effects were also studied; Friedel–Crafts alkylation reactions of indole (**8a**) and *p*-fluoronitrostyrene (**9a**) were also performed using a ligand–metal ratio of 15 mol% of **L5**:Cu(OTf)_2_ at room temperature in several solvents, such as tetrahydrofuran, methanol, acetonitrile, dichloromethane, *n*-hexane and ethylacetate, within various time frames (84–96 h) (Table 1, entries 10–15), where dichloromethane was found to be the best solvent for chemical yield improvement but with no enantioselectivity (Table 1; entry 13), whereas no product formation took place in *n*-hexane and ethylacetate (Table 1, entries 14 and 15), although in THF, moderate yield (48%) and enantioselectivity (55%) were observed (Table 1; entry 10). From the above preliminary findings, it is obvious that a 15 mol% ligand–metal ratio [15 mol% **L5**:Cu(OTf)_2_] in toluene at room temperature in 48 h was the optimum set of reaction conditions to afford the final C–C bond formation adduct. Interestingly, it is clear from the preliminary results that oxazolinyl-based ligands are more potent than imidazolinyl-based ones; more interestingly, the substitution at the oxazolinyl moiety showed to also be critical for the asymmetric induction. Further investigation for better understanding is highly recommended.

Next, another two factors were also investigated, namely metal salts and temperature effects. Therefore, a Friedel–Crafts alkylation of indole (**8a**) with *p-*fluoronitrostyrene (**9a**) was carried out using 15 mol% of ligand **L5** with the combination of several metal triflates, such as Zn(OTf)_2_, Mg(OTf)_2,_ Er(OTf)_2_ and Yb(OTf)_2_, and metal chlorides such as FeCl_3_ and PdCl_2_, in toluene at 25 °C, and the results are summarized in Table 2. It was observed from the metal screening that Zn(OTf)_2_, FeCl_3_ and PdCl_2_ yielded product **10a** with excellent to good chemical yields (97%, 80% and 70%, respectively), while the enantioselectivity remains negligible (Table 2, entries 1, 5 and 6). Two attempts were carried out at low (0 °C) and high (70 °C) temperature for 92 h and 24 h, respectively, and henceforth, 42% and 70% chemical yields with 76% and 65% enantioselectivity were observed (Table 2, entries 7 and 8). The results showed no significant changes for either the chemical yield or the enantioselectivity (Table 2, entry 8). From the overall findings, a catalyst generated in situ from ligand **L5** and Lewis acid Cu(OTf)_2_ in toluene was found to be the optimum reaction condition for the asymmetric Friedel–Crafts alkylation of indole (**8a**) and *p*-fluoronitrostyrene (**9a**).

#### 2.2.2. Substrate Scope

To illustrate the generality, 20 examples of asymmetric Friedel–Crafts alkylation reactions have been carried out using indoles **8a**–**d** with various nitroolefins (**9a**–**h**) under the optimized reaction conditions, i.e., 15 mol% **L5**:Cu(OTf)_2_ in toluene at 25 °C for 48 h, and the results are shown in Table 3. After the observing the results, it seems that substrates **9a**–**h** reacted with indole **8a** moderately and yielded chiral products **10a**–**f** in the range of 40–67% yields with 64–80% enantioselectivity. Substrates **9a**, **9b**, **9d**, **9e and 9h** performed fairly well, yielding corresponding FC products **10a**, **10b**, **10d, 10e** and **10h** with 67, 64, 66, 58 and 60% yields and good enantiomeric excess (*ee*) at 74, 80, 69, 70 and 64% *ee*, respectively (Table 3, entries 1, 2, 4, 5 and 8). While substrates **9c, 9f** and **9g** furnished the corresponding Friedel–Crafts alkylated products **10c, 10f** and **10g** with poor chemical yields (40, 48 and 52%, respectively) because of the steric hindrance of the substrate, the enantioselectivity remained good (75, 71 and 71%, respectively) (Table 3, entries 3, 6 and 7). When substrate **9a**–**h** was allowed to react with 5-bromoindole **(8b)** under the optimized conditions, poor yields were observed (**10i**–**p,** 35–55%) with good enantioselectivity (60–81% *ee*) (Table 3, entries 9–16). A Friedel–Crafts reaction of 5-fluoro indole with *β*-nitrostyrene **9g** furnished a moderate yield (57%) with good enantioselectivity (66% *ee*) as compared to the reaction with the more hindered **9h,** which produced poor yield (45%) as well as poor enantioselectivity (21% *ee*) (Table 3, entries 17 and 18)**.** We further performed the Friedel–Crafts reaction with *N*-ethyl-protected indole and *β*-nitrostyrene **9a** and **9d**, which produced good yields (73 and 76%) with poor enantiomeric excess (35 and 27%) (Table 3, entries 19 and 20). Interestingly, when the asymmetric Friedel–Crafts alkylation of indole **8a** with nitrostyrene **9a** was performed at a large scale (10-fold), both the yield (76%) and enantioselectivity (77% *ee*) were improved (Table 3, entry 1).

Finally, to examine another nitrostyrene system for the Friedel–Crafts arylation, two nitrostyrene (**9i** and **9j**)-based indole scaffold were synthesized and characterized. The synthesized indole-based nitrostyrenes **9i** and **9j** were used as substrates for the asymmetric Friedel–Crafts arylation using our optimized method, but they unfortunately did not succeed in affording the final desired chiral FC products **10u** and **10v**, as shown in Figure 2. The requisite final compounds either did not occur or decomposed.

In Figure 3, the proposed cycle of the catalytic mechanism has been shown, where in the intermediates (**II**) and (**III**), it has been clearly shown that the addition of an incoming nucleophilic group from the *Si* face is more favorable than the *Re* face since the latter is a more sterically hindered face as compared to former.

In case of Friedel–Craft product with indole, the retention time of the *S* enantiomer was found to be lesser than the *R* enantiomer in the chiral HPLC analysis using Daicel OD-H chiral column and *n*-hexane/*iso*-propanol system in the reported literature, while for FC products with 5-bromoindole it was found to be vice versa. Therefore, the absolute configuration of the synthesized chiral FC products **10a**–**d, 10g, 10i**–**l, 10o, 10q, 10s and 10t** was assigned as *S*, while **10e, 10f, 10h, 10m**, **10n, 10p and 10r** were assigned as *R* by comparing their retention time and optical rotation values found in reported literature, assuming that the reaction took place via uniform mechanistic pathway (Table 3) [41,74].

## 3. Materials and Methods

### 3.1. General

Reagents obtained from commercial suppliers were used without further purification. Preparation of *bis*(imidazoline) and *bis*(oxazoline) ligands was performed in dried glassware flasks under a static pressure of nitrogen. Solvents were dried prior to use following standard procedures. Reactions were monitored by thin layer chromatography using Merck silica gel 60 Kieselgel F254 TLC (Merck, Kenilworth, NJ, USA), and column chromatography was performed on silica gel 100–200 (40–63 µm, ASTM) from Merck using the indicated solvents. ^1^H and ^13^C-NMR spectra were recorded in CDCl_3_ and DMSO-*d*_6_ on a *Jeol* Spectrometer (Jeol, Tokyo, Japan) (400 MHz and 500 MHz). The chemical shifts are reported in ppm. All the racemic products were freshly prepared as per the method reported in the literature [83]. Infrared spectra were recorded on a Thermo Scientific Nicolet iS10 FT-IR spectrometer (Thermo Fisher Scientific, Waltham, MA, USA). Enantiomeric ratios were determined by analytical chiral HPLC analysis on a Shimadzu LC-20A (Shimadzu, Kyoto, Japan) Prominence instrument with a chiral stationary phase using Daicel OD-H columns (Chiral Technologies Europe, Illkirch-Graffenstaden, France) and 70–75% *n*-hexane/*iso-*propanol as eluents (Appendix A). Optical rotations were obtained with a PerkinElmer 343 Polarimeter (PerkinElmer, Waltham, MA, USA). Melting points (m.p.) were recorded on a Thomas-Hoover capillary melting point apparatus (Thomas-Hoover, Texas City, USA) and were not corrected. Mass spectrometric analysis was done using ESI mode on an Agilent Technologies 6410-triple quad LC/MS instrument (Agilent, Santa Clara, CA, USA). Elemental analyses were performed on Perkin-Elmer PE 2400 CHN Elemental Analyzer with autosampler, CHN mode. X-ray diffraction data were collected on a Rigaku Oxford Diffraction Supernova diffractometer and processed with CrysAlisPro software v. 1.171.41.93a (Rigaku Oxford Diffraction, Yarnton, UK, 2020) using Cu K_ radiation”.

### 3.2. General Procedure (**GP1**) for the Preparation of Bis(hydroxyamides) **4a–c**

**GP1:** A 100-mL round bottom flask was charged with thiophene-2,5-dicarboxlyic acid (**1**) (0.5 mg, 2.9 mmol) and SOCl_2_ (7 mL). A catalytic amount of DMF (3 drops) was added, and the reaction was reflux for 24 h under inert atmosphere. The reaction was then cooled, and excess SOCl_2_ was removed under reduced pressure to give the corresponding crude acid chloride (**2**). The crude acid chloride **2** (2.9 mmol) solution in CH_2_Cl_2_ (10 mL) was then slowly added to a pre-stirred solution of amino alcohol **3a**–**c** (6.9 mmol, 2.1 eq.) and triethylamine (2 mL, 5 eq.) in CH_2_Cl_2_ (35 mL) at −10 °C. The reaction was then stirred at ambient temperature for 24 h. After reaction completion, the solvents were removed and the residue was poured into water (55 mL). Upon standing at room temperature for 4 h, solid product was precipitated out, which was then collected by filtration and purified by column chromatography using 100–200 mesh silica gel and CH_2_Cl_2_/MeOH (95:5) as an eluent to afford pure products **4a**–**c**.

#### 3.2.1. *N^2^,N^5^*-*Bis*((*S*)-1-Hydroxy-3-methylbutan-2-yl)thiophene-2,5-dicarboxamide (**4a**)

Following **GP1,** thiophene-2,5-dicarboxlyic acid chloride (**2**) and (*S*)-2-amino-3-methylbutan-1-ol (**3a**) reacted to produce 2,5-dicarboxamide alcohol (**4a**) as white solid (0.74 g, 75%); m.p. 199–201 °C; ${\left[\alpha \right]}_{D}^{20}=-26^{o}$(*c* 0.20, CH_3_OH); IR (KBr**,** cm^−1^): 3350, 3086, 3071, 2956, 2870, 2496, 1627, 1543, 1515, 1464, 1033, 743; ^1^H-NMR (400 MHz, DMSO-*d*_6_): *δ*(ppm) = 8.11 (d, *J* = 8.9 Hz, 2H, N**H**), 7.82 (s, 2H, Ar–**H**), 4.63 (t, *J* = 5.8 Hz, 2H, NHC**H**), 3.74 (p, *J* = 7.0, 6.4 Hz, CH_2_O**H**), 3.56–3.45 (m, 4H, C**H**_2_OH), 1.91 (dp, *J* = 13.3, 6.2 Hz, 2H, C**H**(CH_3_)_2_), 0.88 (dd, *J* = 11.5, 6.7 Hz, 12H, CH(C**H**_3_)_2_); ^13^C-NMR (101 MHz, DMSO-*d*6) *δ*(ppm) = 160.8, 143.5, 128.1, 61.2, 56.9, 28.6, 19.6, 18.7; LC/MS (ESI): found 342.2 [M + H]^+^, C_16_H_26_N_2_O_4_S requires 342.16; anal. calcd. for C_16_H_26_N_2_O_4_S: C, 56.12; H, 7.65; N, 8.18; found: C, 55.88; H, 7.72; N, 8.06.

#### 3.2.2. *N^2^,N^5^*-*Bis*((*S*)-1-Hydroxy-4-methylpentan-2-yl)thiophene-2,5-dicarboxamide (**4b**)

Following **GP1,** thiophene-2,5-dicarboxlyic acid chloride (**2**) and (*S*)-2-amino-4-methylpentan-1-ol (**3b**) reacted to produce 2,5-dicarboxamide alcohol (**4b**) as white solid (1.02 g, 95%); m.p. 208–210 °C; ${\left[\alpha \right]}_{D}^{20}=-40.36^{o}$ (*c* 0.11, CH_3_OH); IR (KBr**,** cm^-1^): 3351, 3087, 2958, 2871, 2605, 2498, 1627, 1545, 1517, 1469, 1033, 745; ^1^H-NMR (500 MHz, DMSO-*d*_6_): *δ*(ppm) = 8.19 (d, *J* = 8.7 Hz, 2H, N**H**), 7.77 (s, 2H, Ar–**H**), 4.74 (s, 2H, NHC**H**), 4.04–3.91 (m, 2H, CH_2_O**H**), 3.41 (dt, *J* = 11.0, 5.7 Hz, 2H, C**H**_2_OH), 3.05 (q, *J* = 7.3 Hz, 2H, C**H**_2_OH), 1.66–1.54 (m, 2H, C**H**(CH_3_)_2_), 1.48–1.40 (m, 2H, CHC**H**_2(a)_), 1.38–1.33 (m, 2H, CHC**H**_2(b)_), 0.88 (d, *J* = 6.6 Hz, 6H, CH(C**H**_3_)_2_), 0.86 (d, *J* = 6.6 Hz, 6H, CH(C**H**_3_)_2_). ^13^C-NMR (126 MHz, DMSO-*d*_6_): *δ*(ppm) = 160.5, 143.4, 128.0, 63.8, 49.7, 45.4, 24.4, 23.3, 21.9; LC/MS (ESI): found 371.2 [M + H]^+^, C_18_H_30_N_2_O_4_S requires 370.19; anal. calcd. for C_18_H_30_N_2_O_4_S: C, 58.35; H, 8.16; N, 7.56; found: C, 58.33; H, 8.18; N, 7.55.

#### 3.2.3. *N^2^,N^5^*-*Bis*((2*S*,3*R*)-1-Hydroxy-3-methylpentan-2-yl)thiophene-2,5-dicarboxamide (**4c**)

Following **GP1,** thiophene-2,5-dicarboxlyic acid chloride (**2**) and (2*S*,3*R*)-2-amino-3-methylpentan-1-ol (**3c**) reacted to produce 2,5-dicarboxamide alcohol (**4c**) as white solid (1.04 g, 97%); m.p. 233–234 °C; ${\left[\alpha \right]}_{D}^{20}=-30.39^{o}$ (*c* 0.10, CH_3_OH); IR (KBr**,** cm^-1^): 3352, 3086, 2956, 2870, 2609, 2493, 1625, 1544, 1516, 1465, 1030, 744; ^1^H-NMR (500 MHz, DMSO-*d*_6_): *δ*(ppm) = 8.09 (d, *J* = 8.9 Hz, 2H, N**H**), 7.76 (s, 2H, Ar–**H**), 4.53 (s, 2H, CH_2_O**H**), 3.78–3.70 (m, 2H, NHC**H**), 3.53–3.42 (m, 4H, C**H**_2_OH), 1.68–1.59 (m, 2H, C**H**CH_3_), 1.47–1.37 (m, 2H, C**H**_2_CH_3_), 1.12–1.01 (m, 2H, C**H**_2_CH_3_), 0.83 (d, *J* = 6.9 Hz, 6H, CHC**H**_3_), 0.80 (t, *J* = 7.4 Hz, 6H, CH_2_C**H**_3_); ^13^C-NMR (126 MHz, DMSO-*d*_6_): *δ*(ppm) = 160.7, 143.4, 128.0, 60.9, 55.7, 35.1, 25.1, 15.5, 11.2; LC/MS (ESI): found 399.3 [M + H]^+^, C_20_H_34_N_2_O_4_S requires 398.22; anal. calcd. for C_18_H_30_N_2_O_4_S: C, 58.35; H, 8.16; N, 7.56; found: C, 58.17; H, 8.26; N, 7.44.

### 3.3. General Procedure (**GP2**) for the Preparation of Thiophene-2,5-bis-imidazoline Chiral Ligands (**L1****–L3**)

**GP2:** Thiophene-2,5-dicarboxamide alcohol (**4a**) (1.0 g, 2.92 mmol) in SOC1_2_ (8.76 mL) was refluxed for 24 h. After removal of SOCl_2_, ice-water was added to the residue and the product was extracted with CH_2_Cl_2_ (3 × 25 mL). The combined extracts were washed with brine and dried over anhydrous Na_2_SO_4_. The organics were evaporated to give the crude thiophene-2,5-dicarboxamid dichloride (**5a**). The crude dichloride (**5a**) was then dissolved in dry diethyl ether (20 mL) and the insoluble impurities were filtered out. To this solution, dry triethylamine (4.9 mL, 35.0 mmol, 12.0 eq.) was added, followed by arylamine (**6a**–**c**) (2.5 eq.). After stirring for 12 h at room temperature, 15% NaOH (15 mL) was added and stirred for another 24 h. The aqueous portion was extracted with dichloromethane (3 × 20 mL) and then washed with brine. The combined organics were dried over anhydrous Na_2_SO_4_ and concentrated under reduced pressure to afford crude thiophene-2,5-*bis*(imidazolinyl)thiophene ligands (**L1–L3**). The pure ligands (**L1–L3**) were isolated by column chromatography, using the combination of ethylacetate/petroleumether/Et_3_N (*v:v:v* = 75:24:1) as an eluent.

#### 3.3.1. 2,5-*Bis*((*S*)-4-IsoPropyl-1-phenyl-4,5-dihydro-1*H*-imidazol-2-yl)thiophene (**L1**)

Thiophene-2,5-dicarboxamide alcohol **4a** (1.0 g, 2.92 mmol) and aniline **6a** (0.68 g, 7.3 mmol) were reacted according to **GP2** and afforded yellow-colored ligand **L1** (yield 533 mg, 40%); ${\left[\alpha \right]}_{D}^{20}=+86.24^{o}$ (*c* 0.106, EtOH); ^1^H-NMR (400 MHz, DMSO-*d*_6_): *δ*(ppm) = 7.27 (t, *J* = 7.7 Hz, 4H, Ar–**H**), 7.10 (t, *J* = 7.3 Hz, 2H, Ar–**H**), 6.97 (d, *J* = 8.1 Hz, 4H, Ar–**H**), 6.55 (s, 2H, Ar–**H**), 4.00–3.86 (m, 4H, NC**H**_2_), 3.51 (t, *J* = 8.1 Hz, 2H, NC**H**), 1.72 (p, *J* = 6.6 Hz, 2H, C**H**CH_3_), 0.94 (d, *J* = 7.3 Hz, 6H, CHC**H**_3(a)_), 0.86 (d, *J* = 7.3 Hz, 6H, CHC**H**_3(b)_); ^13^C-NMR (101 MHz, DMSO-*d*_6_): *δ*(ppm) = 154.5, 143.1, 135.2, 129.1, 128.6 124.7, 124.2, 70.13, 59.8, 57.5, 32.7, 18.6; LC/MS (ESI): found 457.2 [M + H]^+^, C_28_H_32_N_4_S requires 456.65; anal. calcd. for C_28_H_32_N_4_S: C, 73.65; H, 7.06; N, 12.27; found: C, 73.60; H, 7.04; N, 12.25.

#### 3.3.2. 2,5-*Bis*((*S*)-1-(4-Chlorophenyl)-4-isopropyl-4,5-dihydro-1*H*-imidazol-2-yl)thiophene (**L2**)

Thiophene-2,5-dicarboxamide alcohol (**4a**) (1.0 g, 2.92 mmol) and 4-chloroaniline (**6b**) (0.93 g, 7.3 mmol) were reacted according to **GP2** and afforded yellow-colored ligand **L2** (yield 583 mg, 38%); ${\left[\alpha \right]}_{D}^{20}=-153.29^{o}$ (*c* 0.07, CH_2_Cl_2_); ^1^H-NMR (400 MHz, DMSO-*d*_6_): *δ*(ppm) = 7.32 (d, *J* = 8.8 Hz, 4H, Ar–**H**), 6.97 (d, *J* = 8.8 Hz, 4H, Ar–**H**), 6.68 (s, 2H, Ar–**H**), 4.00–3.90 (m, 4H, NC**H**_2_), 3.54 (t, *J* = 7.3 Hz, 2H, NC**H**), 1.73 (h, *J* = 6.6 Hz, 2H, C**H**CH_3_), 0.93 (d, *J* = 6.6 Hz, 6H, CHC**H**_3(a)_), 0.85 (d, *J* = 6.6 Hz, 6H, CHC**H**_3(b)_); ^13^C NMR (101 MHz, DMSO-*d*_6_) *δ*(ppm) = 154.0, 141.7, 134.8, 129.1, 128.6, 125.5, 125.3, 70.0, 57.1, 54.9, 32.6, 18.7; LC/MS (ESI): found 525.2 [M + H]^+^, for C_28_H_30_Cl_2_N_4_S requires 524.16; anal. calcd. for C_28_H_30_Cl_2_N_4_S: C, 63.99; H, 5.75; N, 10.66; found: C, 63.87; H, 5.72; N, 10.61.

#### 3.3.3. 2,5-*Bis*((*S*)-4-IsoPropyl-1-(*p*-tolyl)-4,5-dihydro-1*H*-imidazol-2-yl)thiophene (**L3**)

Thiophene-2,5-dicarboxamide alcohol (**4a)** (1.0 g, 2.92 mmol) and *p-*toluidine **6c** (0.78 g, 7.3 mmol) were reacted according to **GP2** and afforded yellow-colored ligand **L3** (yield 538 mg, 38%); ${\left[\alpha \right]}_{D}^{20}=+92.53^{o}$ (*c* 0.05, EtOH); ^1^H-NMR (400 MHz, DMSO-*d_6_*): *δ*(ppm) = 7.08 (d, *J* = 8.1 Hz, 4H, Ar–**H**), 6.89 (d, *J* = 8.1 Hz, 4H, Ar–**H**), 6.52 (s, 2H, Ar–**H**), 3.93–3.86 (m, 4H, NC**H**_2_), 3.47–3.41 (m, 2H, NC**H**), 2.24 (s, 6H, PhC**H**_3_), 1.73 (q, *J* = 6.6 Hz, 2H, C**H**CH_3_), 0.94 (d, *J* = 7.3 Hz, 6H, CHC**H**_3(a)_), 0.85 (d, *J* = 6.6 Hz, 6H, CHC**H**_3(b)_); ^13^C NMR (101 MHz, DMSO-*d*_6_) *δ*(ppm) 154.8, 140.7, 135.2, 134.3, 129.6, 128.6, 124.5, 70.1, 57.8, 32.8, 20.5, 18.7, 18.1; LC/MS (ESI): found 485.3 [M + H]^+^, for C_30_H_36_N_4_S requires 484.27; anal. calcd. for C_30_H_36_N_4_S: C, 74.34; H, 7.49; N, 11.56; found: C, 74.30; H, 7.48; N, 11.52.

### 3.4. General Procedure (**GP3**) for the Synthesis of Thiophene-2,5-bis-oxazoline Chiral Ligands (**L4 and L5**)

**GP3**: Thiophene-2,5-dicarboxamide alcohol (**4b**–**c**) (2.92 mmol) was added to the solution of CH_2_Cl_2_ (60 mL) and triethylamine (4.0 eq., 1.18 g, 11.7 mmol). Catalytic amounts of DMAP (36 mg, 0.1 eq.) and *p*-tosylchoride (695 mg, 3.65 mmol, 1.25 eq.) were added, and the mixture was stirred at 0 °C to r.t. for 48 h. After completion of the reaction, saturated aqueous ammonium chloride solution (100 mL) was added and stirred for another 10 min at room temperature. The organic layer was extracted with CH_2_Cl_2_ (3 × 25 mL) and washed with saturated aqueous NaHCO_3_ solution (50 mL). The combined organic layers were dried over anhydrous Na_2_SO_4_, and the solvent was evaporated in vacuum to afford crude ligands **(L4** and **L5)**, which was purified by column chromatography (5% CH_2_Cl_2_/CH_3_OH) to afford pure thiophene-2,5-*bis*(oxazolinyl)thiophene ligands (**L4** and **L5**).

#### 3.4.1. 2,5-*Bis*((*S*)-4-isoButyl-4,5-dihydrooxazol-2-yl)thiophene (**L4**)

Following the **GP2**, thiophene-2,5-dicarboxamide (**4b**) (1.08 g, 2.92 mmol) underwent direct ring closure reaction to afford ligand **L4** as white solid (yield 586 mg, 60%); m.p. 48–50 °C; ${\left[\alpha \right]}_{D}^{20}=-46.88^{o}$ (*c* 0.093, CH_3_OH); IR (KBr, cm^-1^): 3104, 2953, 2920, 2870, 2847, 1647, 1533, 1251, 1051, 1019, 944, 829; ^1^H-NMR (500 MHz, DMSO-*d_6_*): *δ*(ppm) = 7.53 (s, 2H, Ar–**H**), 4.54 (dd, *J* = 9.3, 8.0 Hz, 2H, OC**H**_2(a)_), 4.31–4.23 (m, 2H, NC**H**), 3.97 (t, *J* = 8.2 Hz, 2H, OC**H**_2(b)_), 1.76 (dt, *J* = 13.5, 6.7 Hz, 2H, C**H**(CH_3_)_2_), 1.52 (dt, *J* = 13.9, 7.0 Hz, 2H, CHC**H**_2(a)_), 1.35 (dt, *J* = 13.5, 7.2 Hz, 2H, CHC**H**_2(b)_), 0.93 (d, *J* = 3.9 Hz, 6H, CH(C**H**_3_)_2_), 0.91 (d, *J* = 3.7 Hz, 6H, CH(C**H**_3_)_2_); ^13^C-NMR (126 MHz, DMSO-*d_6_*): *δ*(ppm) = 157.1, 133.3, 130.5, 73.3, 64.9, 44.7, 25.0, 22.7, 22.5; LC/MS (ESI): found 335.2 [M + H]^+^, C_18_H_26_N_2_O_2_S requires 334.17; anal. calcd. for C_18_H_26_N_2_O_2_S: C, 64.64; H, 7.84; N, 8.38; found: C, 64.62; H, 7.86; N, 8.34.

#### 3.4.2. 2,5-*Bis*((S)-4-((*S*)-sec-Butyl)-4,5-dihydrooxazol-2-yl)thiophene (**L5**)

Following the **GP2**, thiophene-2,5-dicarboxamide (**4c)** (1.08 g, 2.92 mmol) underwent direct ring closure reaction to afford ligand **L5** as white solid (yield 537 mg, 55%); m.p.: 42–43 °C; ${\left[\alpha \right]}_{D}^{20}=-4.06^{o}$ (*c* 0.081, CH_3_OH); IR (KBr, cm^−1^): 3102, 2954, 2921, 2870, 2845, 1648, 1533, 1251, 1052, 1019, 942, 826; ^1^H-NMR (500 MHz, DMSO-*d_6_*): *δ*(ppm) = 7.53 (s, 2H, Ar–**H**), 4.47–4.41 (m, 2H, NC**H**), 4.18–4.11 (m, 4H, OC**H**_2_), 1.62–1.50 (m, 4H, C**H**_2_C**H**_3_), 1.20–1.11 (m, 2H, C**H**CH_3_), 0.90 (t, *J* = 7.3 Hz, 6H, CHC**H**_3_), 0.80 (d, *J* = 6.7 Hz, 6H, CH_2_C**H**_3_); ^13^C-NMR (126 MHz, DMSO-*d_6_*): *δ*(ppm) = 157.2, 133.2, 130.4, 70.9, 70.2, 38.6, 25.4, 14.4, 11.3; LC/MS (ESI): found 335.2 [M + H]^+^, C_18_H_26_N_2_O_2_S requires 334.17; anal. calcd. for C_18_H_26_N_2_O_2_S: C, 64.64; H, 7.84; N, 8.38; found: C, 64.60; H, 7.84; N, 8.38.

### 3.5. Synthesis of the β-nitrostyrene (**9a–j**)

All the *β*-nitrostyrenes (**9a**–**j**) were synthesized by using well-known methods reported in the literature [84]. An oven-dried round bottom flask (100 mL) was charged with aldehydes (10.0 mmol), nitromethane (3.70 g, 60.0 mmol), piperidine (85 mg, 1.0 mmol) and toluene as solvent (10 mL). Anhydrous FeCl_3_ (16.2 mg, 1.0 mmol) was then added to it. The reaction mixture was reflux gently for 4 h under dry condition, using guard tube. The completion of the reaction was confirmed by TLC, and the reaction mixture was cooled to room temperature. The excess solvent was removed under reduced pressure, and the residue was purified by silica gel (100–200 mesh) column chromatography to afford pure *β*-nitrostyrenes **9a**–**j** as yellow solid product (yield 75–90%).

### 3.6. Synthesis of Racemic Friedal–Crafts Alkylated Product Race-(**10a**–**t**)

The racemic products were synthesized by using the reported method [41,83]. Indole derivatives (0.30 mmol), *β*-nitrostyrenes **9a**–**h** (0.30 mmol), FeCl_3_ (10 mol%) and H_2_O (2 mL) were heated at 80 °C for the appropriate time (24 h). After the completion of the reaction, monitored by thin-layer chromatography (TLC), the product was extracted with ethyl acetate (2 × 20 mL). The combined organic layer was dried over anhydrous sodium sulfate, evaporated under reduced pressure and purified by silica gel (100–200 mesh) column chromatography using 15% ethylacetate/*n*-hexane as eluent to afford the pure racemic Friedal–Crafts alkylated product race-(**10a**–**t**) (yield 85 –90%).

### 3.7. General Procedure (**GP4**) for the Asymmetric Friedal–Crafts Alkylation of Indole to β-nitrostyrene (**10a–t**)

**GP4:** An oven-dried screw-capped vial (8 mL) was charged with ligand **L5** (10 mg, 0.03 mmol, 15 mol%), Cu(OTf)_2_ (11 mg, 0.03 mmol, 15 mol %) and dry toluene (3 mL). The mixture was then stirred at reflux for 2 h. After cooling to room temperature, *β*-nitrostyrene **9a**–**h** (0.2 mmol) and 4A° molecular sieves were added. Then, the mixture was stirred for another 30 min, followed by addition of indole **8a**–**d** (0.2 mmol). The reaction was then left stirring for 48 h at room temperature. The solvent was removed under reduced pressure, and the crude product was isolated by flash column chromatography on silica gel with ethylacetate/*n*-hexane (2:8, *v*/*v*) as eluent to afford pure Friedel–Crafts product (**10a**–**t**) in 35–76% isolated yield with 21–81% enantiomeric excess (*ee*).

#### 3.7.1. (*S*)-3-(1-(4-Fluorophenyl)-2-nitroethyl)-1*H*-indole (**10a**)

Indole **8a** (24 mg, 0.2 mmol) and 4-floronitrostyrene (**9a**) (34 mg, 0.2 mmol) were reacted according to the **GP4** to yield product **10a** as colorless oil (isolated yield 38 mg, 67%). Enantiomeric excess (*ee*) was determined by chiral HPLC (Chiracel OD-H column) (70% *n*-hexane/*i*-PrOH, 1.0 mL/min; *t*_major_ = 24.85 min; *t*_minor_ = 30.28 min; λ = 254 nm); 74.3% *ee*; ${\left[\alpha \right]}_{D}^{20}=+32.98^{o}$ (*c* 0.10, CH_3_OH); [Lit. [74] ${\left[\alpha \right]}_{D}^{20}=+39.9^{o}$ (*c* 0.85, CH_2_Cl_2_)]; ^1^H-NMR (500 MHz, CDCl_3_): *δ*(ppm) = 8.13 (s, 1H, N**H**), 7.47–7.40 (m, 1H, Ar–**H**), 7.35 (s, 1H, Ar–**H**), 7.32–7.28 (m, 2H, Ar–**H**), 7.23 (ddd, *J* = 8.2, 7.0, 1.2 Hz, 1H, Ar–**H**), 7.11 (ddd, *J* = 8.1, 6.9, 1.0 Hz, 1H, Ar–**H**), 7.04–6.96 (m, 3H, Ar–**H**), 5.19 (t, *J* = 8.0 Hz, 1H, C**H**), 5.05 (dd, *J* = 12.5, 7.5 Hz, 1H, C**H**_2(a)_), 4.90 (dd, *J* = 12.5, 8.6 Hz, 1H, C**H**_2(b)_); ^13^C-NMR (126 MHz, CDCl_3_): *δ*(ppm) = 163.1 and 161.18 (C_1_-F, *J*_C-F_ = 246.58 Hz), 136.6, 135.07 and 135.04 (C_4_-F, *J*_C-F_ = 3.15 Hz), 129.50 and 129.44 (C_3_-F, *J*_C-F_ = 7.94 Hz), 126.0, 122.9, 121.6, 120.1, 118.9, 115.99 and 115.82 (C_2_-F, *J*_C-F_ = 21.67 Hz), 114.2, 111.6, 79.6, 41.0. All the analytical data are in accordance with the reported literature [42,74].

#### 3.7.2. (*S*)-3-(1-(3-Bromophenyl)-2-nitroethyl)-1*H*-indole (**10b**)

Indole **8a** (24 mg, 0.2 mmol) and 3-bromonitrostyrene (**9b)** (46 mg, 0.2 mmol) were reacted according to the **GP4** to yield product **10b** as colorless oil (isolated yield 44 mg, 64%). Enantiomeric excess (*ee*) was determined by chiral HPLC (Chiracel OD-H column) (70% *n*-hexane/*i*-PrOH, 1.0 mL/min; *t*_major_ = 27.66 min; *t*_minor_ = 36.16 min; λ = 254 nm); 79.5% *ee*; ${\left[\alpha \right]}_{D}^{20}=+14.41^{o}$ (*c* 0.104, CH_3_OH); [Lit. [74] ${\left[\alpha \right]}_{D}^{20}=+14.7^{o}$ (*c* 1.3, CH_2_Cl_2_]; ^1^H-NMR (500 MHz, CDCl_3_): *δ*(ppm) = 8.12 (s, 1H, N**H**), 7.49–7.32 (m, 4H, Ar–**H**), 7.28–7.08 (m, 4H, Ar–**H**), 6.98 (d, *J* = 2.5 Hz, 1H, Ar–**H**), 5.15 (t, *J* = 8.0 Hz, 1H, C**H**), 5.02 (dd, *J* = 12.8, 7.6 Hz, 1H, C**H**_2(a)_), 4.89 (dd, *J* = 12.0, 7.8 Hz, 1H, C**H**_2(b)_); ^13^C-NMR (126 MHz, CDCl_3_): *δ*(ppm) = 141.7, 136.5, 130.9, 130.9, 130.6, 126.6, 126.0, 123.1, 122.9, 121.7, 120.2, 118.8, 113.6, 111.6, 79.2, 41.2. All the analytical data are in accordance with the reported literature [74,76].

#### 3.7.3. (*S*)-3-(2-Nitro-1-(4-(trifluoromethyl)phenyl)ethyl)-1*H*-indole (**10c**)

Indole **8a** (24 mg, 0.2 mmol) and 4-trifluoromethynitrostyrene **9c** (44 mg, 0.2 mmol) were reacted according to the **GP4** to yield product **10c** as colorless oil (isolated yield 27 mg, 40%). Enantiomeric excess (*ee*) was determined by chiral HPLC (Chiracel OD-H column) (70% *n*-hexane/*i*-PrOH, 1.0 mL/min; *t*_major_ = 32.09 min; *t*_minor_ = 39.93 min; λ = 254 nm); 75.4% *ee*; ${\left[\alpha \right]}_{D}^{20}=+6.93^{o}$ (*c* 0.05, CH_3_OH); [Lit. [75] ${\left[\alpha \right]}_{D}^{20}=+2.9^{o}$ (c 1.0, CHCl_3_]; ^1^H-NMR (500 MHz, CDCl_3_): *δ*(ppm) = 8.17 (s, 1H, N**H**), 7.59 (d, *J* = 8.1 Hz, 2H, Ar–**H**), 7.47 (d, *J* = 8.1 Hz, 2H, Ar–**H**), 7.42 (dq, *J* = 8.0, 1.0 Hz, 1H, Ar–**H**), 7.38 (dt, *J* = 8.3, 0.9 Hz, 1H, Ar–**H**), 7.23 (ddd, *J* = 8.2, 7.1, 1.2 Hz, 1H, Ar–**H**), 7.13–7.08 (m, 1H, Ar–**H**), 7.03 (dd, *J* = 2.6, 0.9 Hz, 1H, Ar–**H**), 5.26 (t, *J* = 8.0 Hz, 1H, C**H**), 5.09 (dd, *J* = 12.8, 7.4 Hz, 1H, C**H**_2(a)_), 4.97 (dd, *J* = 12.7, 8.7 Hz, 1H, C**H**_2(b)_); ^13^C-NMR (126 MHz, CDCl_3_): *δ*(ppm) = 143.4, 136.6, 128.3, 126.11, 126.07, 126.04, 125.9, 123.1, 121.8, 120.3, 118.8, 113.6, 111.7, 79.1, 41.4. All the analytical data are in accordance with the reported literature [75].

#### 3.7.4. (S)-3-(1-(4-Methoxyphenyl)-2-nitroethyl)-1*H*-indole (**10d**)

Indole **8a** (24 mg, 0.2 mmol) and 4-methoxynitrostyrene **9d** (36 mg, 0.2 mmol) were reacted according to the **GP4** to yield product **10d** as white solid (isolated yield 39 mg, 66%), m.p. 148–149 °C; Enantiomeric excess (*ee*) was determined by chiral HPLC (Chiracel OD-H column) (70% *n*-hexane/*i*-PrOH, 1.0 mL/min; *t*_major_ = 26.24 min; *t*_minor_ = 32.20 min; λ = 254 nm); 69.3% *ee*; ${\left[\alpha \right]}_{D}^{20}=+12.13^{o}$ (*c* 0.53, CH_3_OH) [Lit.[74] ${\left[\alpha \right]}_{D}^{20}=+26.4^{o}$ (*c* 1.1, CH_2_Cl_2_)]; ^1^H-NMR (500 MHz, CDCl_3_): *δ*(ppm) = 8.09 (s, 1H, N**H**), 7.44 (d, *J* = 8.0 Hz, 1H, Ar–**H**), 7.36 (dd, *J* = 8.2, 1.0 Hz, 1H, Ar–**H**), 7.29–7.23 (m, 2H, Ar–**H**), 7.20 (tt, *J* = 8.2, 1.2 Hz, 1H, Ar–**H**), 7.12–7.06 (m, 1H, Ar–**H**), 7.02 (dd, *J* = 2.5, 1.1 Hz, 1H, Ar–**H**), 6.91–6.81 (m, 2H, Ar–**H**), 5.14 (t, *J* = 8.0 Hz, 1H, C**H**), 5.05 (dd, *J* = 12.4, 7.5 Hz, 1H, C**H**_2(a)_), 4.90 (dd, *J* = 12.4, 8.5 Hz, 1H, C**H**_2(b)_), 3.78 (s, 3H, C**H**_3_); ^13^C-NMR (126 MHz, CDCl_3_): *δ*(ppm) = 159.0, 136.6, 131.3, 129.0, 126.2, 122.8, 121.6, 120.1, 119.1, 114.9, 114.4, 111.5, 79.9, 55.4, 41.0. All the analytical data are in accordance with the reported literature [42,74].

#### 3.7.5. (*R*)-3-(2-Nitro-1-(2-nitrophenyl)ethyl)-1*H*-indole (**10e**)

Indole **8a** (24 mg, 0.2 mmol) and 2-nitronitrostyrene **9e** (39 mg, 0.2 mmol) were reacted according to the **GP4** to yield product **10d** as yellow oil (isolated yield 36 mg, 58%). Enantiomeric excess (*ee*) was determined by chiral HPLC (Chiracel OD-H column) (70% *n*-hexane/*i*-PrOH, 1.0 mL/min; *t*_minor_ = 37.48 min; *t*_major_ = 67.98 min; λ = 254 nm); 70.0% *ee*; ${\left[\alpha \right]}_{D}^{20}=+95.57^{o}$ (*c* 0.053, CH_3_OH); [Lit. [80] ${\left[\alpha \right]}_{D}^{20}=+55.3^{o}$ (c 0.7, CH_2_Cl_2_)]; ^1^H-NMR (500 MHz, CDCl_3_): *δ*(ppm) = 8.23 (s, 1H, N**H**), 7.90 (dd, *J* = 8.2, 1.4 Hz, 1H, Ar–**H**), 7.48 (td, *J* = 7.6, 1.4 Hz, 1H, Ar–**H**), 7.43 (dd, *J* = 7.9, 1.6 Hz, 1H, Ar–**H**), 7.39 (ddd, *J* = 8.5, 7.3, 1.6 Hz, 1H, Ar–**H**), 7.35–7.30 (m, 2H, Ar–**H**), 7.21–7.16 (m, 1H, Ar–**H**), 7.12 (d, *J* = 2.6 Hz, 1H, Ar–**H**), 7.07–7.02 (m, 1H, Ar–**H**), 5.88 (t, *J* = 7.7 Hz, 1H, C**H**), 5.12 (dd, *J* = 13.2, 7.1 Hz, 1H, C**H**_2(a)_), 5.07 (dd, *J* = 13.2, 8.3 Hz, 1H, C**H**_2(b)_); ^13^C-NMR (126 MHz, CDCl_3_): *δ*(ppm) = 149.7, 136.5, 133.8, 133.4, 130.1, 128.7, 126.0, 125.2, 123.0, 122.2, 120.3, 118.7, 112.8, 111.6, 78.2, 36.5. All the analytical data are in accordance with the reported literature [80].

#### 3.7.6. (*R*)-3-(1-(2,4-Dichlorophenyl)-2-nitroethyl)-1*H*-indole (**10f**)

Indole **8a** (24 mg, 0.2 mmol) and 2,4-dichloronitronitrostyrene **9f** (44 mg, 0.2 mmol) were reacted according to the **GP4** to yield product **10d** as yellow oil (isolated yield 32 mg, 48%). Enantiomeric excess (*ee*) was determined by chiral HPLC (Chiracel OD-H column) (70% *n*-hexane/*i*-PrOH, 1.0 mL/min; *t*_minor_ = 21.25 min; *t*_major_ = 35.78 min; λ = 254 nm); 71.25% *ee*; ${\left[\alpha \right]}_{D}^{20}=+38.22^{o}$ (*c* 0.052, CH_3_OH); [Lit. [74] ${\left[\alpha \right]}_{D}^{20}=+59.5^{o}$ (c 0.8, CH_2_Cl_2_)]; ^1^H-NMR (500 MHz, CDCl_3_): *δ*(ppm) = 8.16 (s, 1H, N**H**), 7.47 (t, *J* = 1.3 Hz, 1H, Ar-**H**), 7.38 (ddt, *J* = 14.8, 8.2, 0.9 Hz, 2H, Ar–**H**), 7.24–7.20 (m, 1H, Ar–**H**), 7.14 (d, *J* = 1.2 Hz, 2H, Ar–**H**), 7.12–7.08 (m, 2H, Ar–**H**), 5.71–5.66 (m, 1H, C**H**), 4.99 (dd, *J* = 12.9, 8.7 Hz, 1H, C**H**_2(a)_), 4.93 (dd, *J* = 12.9, 7.0 Hz, 1H, C**H**_2(b)_); ^13^C-NMR (126 MHz, CDCl_3_): *δ*(ppm) = 136.6, 135.3, 134.7, 134.2, 130.1, 130.0, 127.8, 126.1, 123.1, 122.0, 120.3, 118.9, 113.0, 111.6, 77.6, 37.7. All the analytical data are in accordance with the reported literature [41,74].

#### 3.7.7. (*S*)-3-(2-Nitro-1-(thiophen-2-yl)ethyl)-1*H*-indole (**10g**)

Indole **8a** (24 mg, 0.2 mmol) and (E)-2-(2-nitrovinyl)thiophene **9g** (31 mg, 0.2 mmol) were reacted according to the **GP4** to yield product **10n** as brown oil (isolated yield 28 mg, 52%). Enantiomeric excess (*ee*) was determined by chiral HPLC (Chiracel OD-H column) (75% *n*-hexane/*i*-PrOH, 1.0 mL/min; *t*_minor_ = 28.57 min; *t*_major_ = 32.38 min; λ = 254 nm); 71.3% *ee*; ${\left[\alpha \right]}_{D}^{20}=+32.18^{o}$ (*c* 0.037, CH_3_OH); ^1^H-NMR (500 MHz, CDCl_3_): *δ*(ppm) = 8.15 (s, 1H, N**H**), 7.53 (d, *J* = 8.0 Hz, 1H, Ar–**H**), 7.37 (d, *J* = 8.2 Hz, 1H, Ar–**H**), 7.25–7.21 (m, 1H, Ar–**H**), 7.20–7.18 (m, 1H, Ar–**H**), 7.15–7.10 (m, 1H, Ar–**H**), 7.09 (d, *J* = 2.58 Hz, 1H, Ar–**H**), 7.01–6.99 (m, 1H, Ar–**H**), 6.95 (dd, *J* = 5.1, 3.6 Hz, 1H, Ar–**H**), 5.47 (t, *J* = 7.9 Hz, 1H, C**H**), 5.08–4.96 (m, 2H, C**H**_2_); ^13^C-NMR (126 MHz, CDCl_3_): *δ*(ppm) = 143.07, 136.53, 127.08, 125.84, 125.38, 125.03, 122.88, 122.09, 120.20, 118.93, 114.15, 111.65, 80.13, 37.05. All the analytical data are in accordance with the reported literature [42].

#### 3.7.8. (*R*)-3-(1-(2,6-Dichlorophenyl)-2-nitroethyl)-1*H*-indole (**10h**)

Indole **8a** (24 mg, 0.2 mmol) and 2,6-dichloronitronitrostyrene **9h** (44 mg, 0.2 mmol) were reacted according to the **GP4** to yield product **10q** as brown oil (isolated yield 40 mg, 60%). Enantiomeric excess (*ee*) was determined by chiral HPLC (Chiracel OD-H column) (75% *n*-hexane/*i*-PrOH, 1.0 mL/min; *t*_minor_ = 11.59 min; *t*_major_ = 13.27 min; λ = 254 nm); 64.1% *ee*; ${\left[\alpha \right]}_{D}^{20}=+91.76^{o}$ (*c* 0.031, CH_3_OH); ^1^H-NMR (500 MHz, CDCl_3_): *δ*(ppm) = 8.14 (s, 1H, N**H**), 7.42 (dq, *J* = 8.0, 0.9 Hz, 1H, Ar–**H**), 7.37–7.23 (m, 3H, Ar–**H**), 7.21–7.13 (m, 3H, Ar–**H**), 7.06 (ddd, *J* = 8.0, 7.1, 1.0 Hz, 1H, Ar–**H**), 6.21 (ddd, *J* = 8.4, 7.4, 1.2 Hz, 1H, C**H**), 5.43 (dd, *J* = 12.8, 7.4 Hz, 1H, C**H**_2(a)_), 5.36 (dd, *J* = 12.8, 8.0 Hz, 1H, C**H**_2(b)_); ^13^C-NMR (126 MHz, CDCl_3_): *δ*(ppm) = 136.17, 134.32, 130.47, 129.90, 129.48, 126.46, 122.71, 122.65, 120.17, 119.06, 111.63, 111.44, 76.44, 38.03. All the analytical data are in accordance with the reported literature [41].

#### 3.7.9. (*S*)-5-Bromo-3-(1-(4-fluorophenyl)-2-nitroethyl)-1*H*-indole (**10i**)

5-bromoindole **8b** (39 mg, 0.2 mmol) and 4-floronitrostyrene **9a** (34 mg, 0.2 mmol) were reacted according to the **GP4** to yield product **10g** as yellow oil (isolated yield 40 mg, 55%). Enantiomeric excess (*ee*) was determined by chiral HPLC (Chiracel OD-H column) (70% *n*-hexane/i-PrOH, 1.0 mL/min; *t*_minor_ = 9.58 min; *t*_major_ = 14.14 min; λ = 254 nm); 77.2% *ee*; ${\left[\alpha \right]}_{D}^{20}=-20.93^{o}$ (*c* 0.051, CH_3_OH); IR (KBr): 3417, 1544, 1376, 1242, 1179, 1028, 743, 549, 524 cm^-1^; ^1^H-NMR (500 MHz, CDCl_3_): *δ*(ppm) = 8.28 (s, 1H, N**H**), 7.55 (d, *J* = 1.8 Hz, 1H, Ar–**H**), 7.34–7.29 (m, 3H, Ar–**H**), 7.25 (d, *J* = 8.6 Hz, 1H, Ar–**H**), 7.09–7.03 (m, 3H, Ar–**H**), 5.14 (t, *J* = 8.0 Hz, 1H, C**H**), 5.04 (dd, *J* = 12.6, 7.8 Hz, 1H, C**H**_2(a)_), 4.91 (dd, *J* = 12.5, 8.2 Hz, 1H, C**H**_2(b)_); ^13^C-NMR (126 MHz, CDCl_3_) *δ*(ppm) = 163.23 and 161.27 (C_1_-F, *J*_C-F_ = 247.21 Hz), 135.2, 134.58 and 134.55 (C_4_-F, *J*_C-F_ = 3.28 Hz), 129.44 and 129.37 (C_3_-F, *J*_C-F_ = 8.19 Hz), 127.8, 125.9, 122.8, 121.5, 116.17 and 116.00 (C_2_-F, *J*_C-F_ = 21.55 Hz), 113.9, 113.4, 113.1, 79.5, 40.7; LC/MS (ESI): found 363.02 [M+H]^+^, C_16_H_12_BrFN_2_O_2_ requires 362.01; anal. calcd. for C_16_H_12_BrFN_2_O_2_: C, 52.91; H, 3.33; N, 7.71; found: C, 53.01; H, 3.39; N, 7.65.

#### 3.7.10. (*S*)-5-Bromo-3-(1-(3-bromophenyl)-2-nitroethyl)-1*H*-indole (**10j**)

5-bromoindole **8b** (39 mg, 0.2 mmol) and 3-bromonitrostyrene **9b** (46 mg, 0.2 mmol) were reacted according to the **GP4** to yield product **10h** as yellow oil (isolated yield 39 mg, 46%). Enantiomeric excess (*ee*) was determined by chiral HPLC (Chiracel OD-H column) (80% *n*-hexane/*i*-PrOH, 1.0 mL/min; *t*_minor_ = 21.41 min; *t*_major_ = 34.37 min; λ = 254 nm); 79.5% *ee*; ${\left[\alpha \right]}_{D}^{20}=-45.13^{o}$ (*c* 0.053, CH_3_OH); IR (KBr): 3401, 1538, 1378, 1009, 814, 745, 589, 535, 421 cm^-1^; ^1^H-NMR (500 MHz, CDCl_3_): *δ*(ppm) = 8.24 (s, 1H, N**H**), 7.52 (d, *J* = 1.9 Hz, 1H, Ar–**H**), 7.42–7.37 (m, 2H, Ar–**H**), 7.28–7.22 (m, 2H, Ar–**H**), 7.21–7.16 (m, 2H, Ar–**H**), 7.03 (dd, *J* = 2.6, 0.9 Hz, 1H, Ar–**H**), 5.07 (t, *J* = 8.0 Hz, 1H, C**H**), 4.97 (dd, *J* = 12.7, 8.0 Hz, 1H, C**H**_2(a)_), 4.86 (dd, *J* = 12.7, 8.0 Hz, 1H, C**H**_2(b)_); ^13^C-NMR (126 MHz, CDCl_3_): *δ*(ppm) = 141.2, 135.2, 131.1, 130.8, 130.7, 127.8, 126.5, 126.0, 123.2, 122.9, 121.3, 113.5, 113.3, 113.1, 79.2, 41.0; LC/MS (ESI): found 423.01 [M+H]^+^, C_16_H_12_Br_2_N_2_O_2_ requires 421.93; anal. calcd. for C_16_H_12_Br_2_N_2_O_2_: C, 45.31; H, 2.85; N, 6.61; found: C, 45.23; H, 2.96; N, 6.52.

#### 3.7.11. (*S*)-5-Bromo-3-(2-nitro-1-(4-(trifluoromethyl)phenyl)ethyl)-1*H*-indole (**10k**)

5-bromoindole **8b** (39 mg, 0.2 mmol) and 4-trifluoromethynitrostyrene **9c** (44 mg, 0.2 mmol) were reacted according to the **GP4** to yield product **10i** as colorless oil (isolated yield 29 mg, 35%). Enantiomeric excess (*ee*) was determined by chiral HPLC (Chiracel OD-H column) (75% *n*-hexane/*i*-PrOH, 1.0 mL/min; *t*_minor_ = 11.80 min; *t*_major_ = 19.82 min; λ = 254 nm); 78.43% *ee*; ${\left[\alpha \right]}_{D}^{20}=-29.51^{o}$ (*c* 0.056, CH_3_OH); IR (KBr): 3418, 1537, 1371, 1247, 1103, 715, 519 cm^-1^; ^1^H-NMR (500 MHz, CDCl_3_): *δ*(ppm) = 8.22 (s, 1H, N**H**), 7.60 (d, *J* = 8.1 Hz, 2H, Ar–**H**), 7.53 (d, *J* = 1.8 Hz, 1H, Ar–**H**), 7.44 (d, *J* = 8.1 Hz, 2H, Ar–**H**), 7.30 (dd, *J* = 8.7, 1.9 Hz, 1H, Ar–**H**), 7.25 (d, *J* = 8.7 Hz, 1H, Ar–**H**), 7.07 (d, *J* = 2.6 Hz, 1H, Ar–**H**), 5.20 (t, *J* = 8.0 Hz, 1H, C**H**), 5.04 (dd, *J* = 12.8, 7.6 Hz, 1H, C**H**_2(a)_), 4.94 (dd, *J* = 12.8, 8.4 Hz, 1H, C**H**_2(b)_); ^13^C-NMR (126 MHz, CDCl_3_): *δ*(ppm) = 142.9, 135.2, 128.2, 127.7, 126.3, 126.23, 126.20, 126.1, 122.9, 121.4, 113.7, 113.3, 113.2, 79.0, 41.1; LC/MS (ESI): found 423.01 [M+H]^+^, C_17_H_12_BrF_3_N_2_O_2_ requires 421.93; anal. calcd. for C_17_H_12_BrF_3_N_2_O_2_: C, 49.42; H, 2.93; N, 6.78; found: C, 49.61; H, 3.07; N, 6.69.

#### 3.7.12. (*S*)-5-Bromo-3-(1-(4-methoxyphenyl)-2-nitroethyl)-1*H*-indole (**10l**)

5-bromoindole **8b** (39 mg, 0.2 mmol) and 4-methoxynitrostyrene **9d** (36 mg, 0.2 mmol) were reacted according to the **GP4** to yield product **10j** as white solid (isolated yield 29 mg, 39%), m.p. 145–146 °C; Enantiomeric excess (*ee*) was determined by chiral HPLC (Chiracel OD-H column) (75% *n*-hexane/*i*-PrOH, 1.0 mL/min; *t*_minor_ = 17.33 min; *t*_major_ = 20.21 min; λ = 254 nm; 62.6% *ee*; ${\left[\alpha \right]}_{D}^{20}=-29.43^{o}$ (*c* 0.053, CH_3_OH); ^1^H-NMR (500 MHz, CDCl_3_): *δ*(ppm) = 8.13 (s, 1H, N**H**), 7.53 (d, *J* = 1.9 Hz, 1H, Ar–**H**), 7.26 (d, *J* = 3.4 Hz, 1H, Ar–**H**), 7.23–7.18 (m, 3H, Ar–**H**), 7.06 (dd, *J* = 2.6, 0.9 Hz, 1H, Ar–-**H**), 6.89–6.83 (m, 2H, Ar–**H**), 5.07 (t, *J* = 8.0 Hz, 1H, C**H**), 4.99 (dd, *J* = 12.3, 8.0 Hz, 1H, C**H**_2(a)_), 4.87 (dd, *J* = 12.3, 8.0 Hz, 1H, C**H**_2(b)_), 3.78 (s, 3H, C**H**_3_); ^13^C-NMR (126 MHz, CDCl_3_): *δ*(ppm) = 159.2, 135.3, 130.8, 128.9, 128.0, 125.8, 122.7, 121.7, 114.6, 114.5, 113.4, 112.9, 79.7, 55.4, 40.7. All the analytical data are in accordance with the reported literature [81].

#### 3.7.13. (*R*)-5-Bromo-3-(2-nitro-1-(2-nitrophenyl)ethyl)-1*H*-indole (**10m**)

5-bromoindole **8b** (39 mg, 0.2 mmol) and 2-nitronitrostyrene **9e** (39 mg, 0.2 mmol) were reacted according to the **GP4** to yield product **10k** as yellow oil (isolated yield 33 mg, 42%). Enantiomeric excess (*ee*) was determined by chiral HPLC (Chiracel OD-H column) (75% *n*-hexane/*i*-PrOH, 1.0 mL/min; *t*_minor_ = 25.65 min; *t*_major_ = 28.75 min; λ = 254 nm); 77.69% *ee*; ${\left[\alpha \right]}_{D}^{20}=+21.38^{o}$ (*c* 0.07, CH_3_OH); IR (KBr): 3419, 1548, 1513, 1339, 723, 431 cm^-1^; ^1^H-NMR (500 MHz, CDCl_3_): *δ*(ppm) = 8.32 (s, 1H, N**H**), 7.92 (dd, *J* = 8.1, 1.4 Hz, 1H, Ar–**H**), 7.55–7.49 (m, 1H, Ar–**H**), 7.46–7.38 (m, 3H, Ar–**H**), 7.27–7.23 (m, 1H, Ar–**H**), 7.20 (d, *J* = 8.6 Hz, 1H, Ar–**H**), 7.13 (d, *J* = 2.6 Hz, 1H, Ar–**H**), 5.83 (t, *J* = 7.7 Hz, 1H, C**H**), 5.10 (dd, *J* = 13.3, 7.0 Hz, 1H, C**H**_2(a)_), 5.03 (dd, *J* = 13.3, 8.4 Hz, 1H, C**H**_2(b)_); ^13^C-NMR (126 MHz, CDCl_3_): *δ*(ppm) = 149.6, 135.2, 133.5, 133.4, 129.9, 129.0, 127.7, 126.1, 125.5, 123.5, 121.3, 113.6, 113.1, 112.3, 78.1, 36.4; LC/MS (ESI): found 390.02 [M+H]^+^, C_16_H_12_BrN_3_O_4_ requires 389.00; anal. calcd. for C_16_H_12_BrN_3_O_4_: C, 49.25; H, 3.10; N, 10.77; found: C, 49.33; H, 3.17; N, 10.84.

#### 3.7.14. (*R*)-5-Bromo-3-(1-(2,4-dichlorophenyl)-2-nitroethyl)-1*H*-indole (**10n**)

5-bromoindole **8b** (39 mg, 0.2 mmol) and 2,4-dichloronitronitrostyrene **9f** (44 mg, 0.2 mmol) were reacted according to the **GP4** to yield product **10l** as brown oil (isolated yield 31 mg, 37%). Enantiomeric excess (*ee*) was determined by chiral HPLC (Chiracel OD-H column) (75% *n*-hexane/*i*-PrOH, 1.0 mL/min; *t*_minor_ = 9.90 min; *t*_major_ = 20.31 min; λ = 254 nm); 74.6% *ee*; ${\left[\alpha \right]}_{D}^{20}=-22.88^{o}$ (*c* 0.056, CH_3_OH); IR (KBr): 3417, 1542, 1456, 1348, 1098, 809, 742, 587 cm^−1^; ^1^H-NMR (500 MHz, CDCl_3_): *δ*(ppm) = 8.26 (s, 1H, N**H**), 7.48 (dd, *J* = 18.0, 2.0 Hz, 2H, Ar–**H**), 7.28–7.24 (m, 1H, Ar–**H**), 7.21 (dd, *J* = 8.6, 1.0 Hz, 1H, Ar–**H**), 7.14 (dd, *J* = 8.4, 2.1 Hz, 1H, Ar–**H**), 7.10 (dd, *J* = 2.6, 1.1 Hz, 1H, Ar–**H**), 7.07 (dd, *J* = 8.4, 1.1 Hz, 1H, Ar–**H**), 5.59 (t, *J* = 7.9 Hz, 1H, C**H**), 4.92 (d, *J* = 1.9 Hz, 1H, C**H**_2(a)_), 4.91 (d, *J* = 1.1 Hz, 1H, C**H**_2(b)_); ^13^C-NMR (126 MHz, CDCl_3_): *δ*(ppm) = 135.2, 134.8, 134.6, 134.4, 130.2, 129.8, 127.8, 126.1, 123.3, 121.4, 113.6, 113.1, 112.5, 77.4, 37.5; LC/MS (ESI): found 413.01 [M+H]^+^, C_16_H_11_BrCl_2_N_2_O_2_ requires 411.94; anal. calcd. for C_16_H_11_BrCl_2_N_2_O_2_: C, 46.41; H, 2.68; N, 6.77; found: C, 46.27; H, 2.57; N, 6.79.

#### 3.7.15. (*S*)-5-Bromo-3-(2-nitro-1-(thiophen-2-yl)ethyl)-1*H*-indole (**10o**)

5-bromoindole **8b** (39 mg, 0.2 mmol) and (E)-2-(2-nitrovinyl)thiophene **9g** (31 mg, 0.2 mmol) were reacted according to the **GP4** to yield product **10m** as brown oil (isolated yield 33 mg, 47%). Enantiomeric excess (*ee*) was determined by chiral HPLC (Chiracel OD-H column) (75% *n*-hexane/*i*-PrOH, 1.0 mL/min; *t*_minor_ = 12.18 min; *t*_major_ = 20.57 min; λ = 254 nm); 72.0% *ee*; ${\left[\alpha \right]}_{D}^{20}=-6.87^{o}$ (*c* 0.081, CH_3_OH); ^1^H-NMR (500 MHz, CDCl_3_): *δ*(ppm) = 8.50 (s, 1H, N**H**), 7.62 (d, *J* = 2.0 Hz, 1H, Ar–**H**), 7.27 (dd, *J* = 8.7, 1.9 Hz, 1H, Ar–**H**), 7.22–7.18 (m, 2H, Ar–**H**), 7.10 (d, *J* = 2.6 Hz, 1H, Ar–**H**), 6.97–6.92 (m, 2H, Ar–**H**), 5.38 (t, *J* = 7.9 Hz, 1H, C**H**), 5.03–4.93 (m, 2H, C**H**_2_); ^13^C-NMR (126 MHz, CDCl_3_): *δ*(ppm) = 142.57, 135.16, 127.56, 127.18, 125.72, 125.46, 125.20, 123.31, 121.41, 113.62, 113.39, 113.16, 79.95, 36.80. All the analytical data are in accordance with the reported literature [42].

#### 3.7.16. (*R*)-5-Bromo-3-(1-(2,6-dichlorophenyl)-2-nitroethyl)-1*H*-indole (**10p**)

5-Bromoindole **8b** (39 mg, 0.2 mmol) and 2,6-dichloronitronitrostyrene **9h** (44 mg, 0.2 mmol) were reacted according to the **GP4** to yield product **10p** as brown oil (isolated yield 43 mg, 52%). Enantiomeric excess (*ee*) was determined by chiral HPLC (Chiracel OD-H column) (75% *n*-hexane/*i*-PrOH, 1.0 mL/min; *t*_minor_ = 10.29 min; *t*_major_ = 11.29 min; λ = 254 nm); 60.1% *ee*; ${\left[\alpha \right]}_{D}^{20}=+32.64^{o}$ (*c* 0.034, CH_3_OH); IR (KBr): 3415, 1549, 1463, 1356, 1109, 822, 734, 605, 541, 424 cm^−1^; ^1^H-NMR (500 MHz, CDCl_3_): *δ*(ppm) = 8.85 (s, 1H, N**H**), 7.50 (s, 1H, Ar–**H**), 7.32 (s, 1H, Ar–**H**), 7.26 (s, 1H, Ar–**H**), 7.22–7.19 (m, 2H, Ar–**H**), 7.18–7.13 (m, 2H, Ar–**H**), 6.12 (td, *J* = 7.7, 1.2 Hz, 1H, C**H**), 5.39 (dd, *J* = 12.9, 7.7 Hz, 1H, C**H**_2(a)_), 5.29 (dd, *J* = 12.9, 7.8 Hz, 1H, C**H**_2(b)_); ^13^C-NMR (126 MHz, CDCl_3_): *δ*(ppm) = 134.86, 133.93, 130.06129.64, 128.19, 125.38, 124.10, 121.60, 114.17, 113.25, 112.94, 111.02, 76.27, 37.77; LC/MS (ESI): found 412.98 [M+H]^+^, C_16_H_11_BrCl_2_N_2_O_2_ requires 411.94; Anal. calcd. for C_16_H_11_BrCl_2_N_2_O_2_: C, 46.41; H, 2.68; N, 6.77; Found: C, 46.36; H, 2.74; N, 6.63.

#### 3.7.17. (*S*)-5-Fluoro-3-(2-nitro-1-(thiophen-2-yl)ethyl)-1*H*-indole(**10q**)

5-Fluoroindole **8c** (27 mg, 0.2 mmol) and (E)-2-(2-nitrovinyl)thiophene **9g** (31 mg, 0.2 mmol) were reacted according to the **GP4** to yield product **10o** as brown oil (isolated yield 33 mg, 57%). Enantiomeric excess (*ee*) was determined by chiral HPLC (Chiracel OD-H column) (75% *n*-hexane/*i*-PrOH, 1.0 mL/min; *t*_minor_ = 11.81 min; *t*_major_ = 13.42 min; λ = 254 nm); 66.0% *ee*; ${\left[\alpha \right]}_{D}^{20}=+36.97^{o}$ (*c* 0.035, CH_3_OH); IR (KBr): 3417, 1547, 1469, 1343, 1205, 827, 731, 541 cm^-1^; ^1^H-NMR (500 MHz, CDCl_3_): *δ*(ppm) = 8.18 (s, 1H, N**H**), 7.40 (dd, *J* = 8.7, 5.2 Hz, 1H, Ar–**H**), 7.21 (dd, *J* = 5.1, 1.3 Hz, 1H, Ar–**H**), 7.13–7.08 (m, 1H, Ar–**H**), 7.04 (dd, *J* = 9.4, 2.3 Hz, 1H, Ar–**H**), 7.00–6.92 (m, 2H, Ar–**H**), 6.87 (td, *J* = 9.2, 2.3 Hz, 1H, Ar–**H**), 5.43 (t, *J* = 7.9 Hz, 1H, C**H**), 5.05–4.96 (m, 2H, C**H**_2_); ^13^C-NMR (126 MHz, CDCl_3_): *δ*(ppm) = 161.31 and 159.41 (C_1_-F, *J*_C-F_ = 239.40 Hz), 142.84, 136.59, 136.49, 127.15, 125.46, 125.18, 122.48, 122.28 and 122.25 (C_4_-F, *J*_C-F_ = 3.53 Hz), 119.84 and 119.76 (C_3_-F, *J*_C-F_ = 10.04 Hz), 114.35, 109.22 and 109.02 (C_2_-F, *J*_C-F_ = 23.94 Hz), 80.09, 36.99; LC/MS (ESI): found 291.10 [M+H]^+^, C_14_H_11_FN_2_O_2_S requires 290.05; Anal. calcd. for C_14_H_11_FN_2_O_2_S: C, 57.92; H, 3.82; N, 9.65; Found: C, 58.11; H, 3.93; N, 9.52.

#### 3.7.18. (*R*)-3-(1-(2,6-Dichlorophenyl)-2-nitroethyl)-1*H*-indole (**10r**)

5-Fluoroindole **8c** (27 mg, 0.2 mmol) and 2,6-dichloronitronitrostyrene **9h** (44 mg, 0.2 mmol) were reacted according to the **GP4** to yield product **10r** as brown oil (isolated yield 32 mg, 45%). Enantiomeric excess (*ee*) was determined by chiral HPLC (Chiracel OD-H column) (75% *n*-hexane/*i*-PrOH, 1.0 mL/min; *t*_minor_ = 7.99 min; *t*_major_ = 10.29 min; λ = 254 nm); 24.3% *ee*; ${\left[\alpha \right]}_{D}^{20}=+65.97$ (*c* 0.029, CH_3_OH); IR (KBr): 3418, 1551, 1472, 1371, 1101, 819, 735 cm^−1^; ^1^H-NMR (500 MHz, CDCl_3_): *δ*(ppm) = 8.20 (s, 1H, N**H**), 7.43–7.22 (m, 3H, Ar–**H**), 7.18–7.12 (m, 2H, Ar–**H**), 7.02 (dd, *J* = 9.4, 2.3 Hz, 1H, Ar–**H**), 6.81 (ddd, *J* = 9.5, 8.8, 2.3 Hz, 1H, Ar–**H**), 6.17 (td, *J* = 7.6, 1.2 Hz, 1H, C**H**), 5.42 (dd, *J* = 12.8, 7.6 Hz, 1H, C**H**_2(a)_), 5.31 (dd, *J* = 12.9, 7.7 Hz, 1H, C**H**_2(b)_); ^13^C-NMR (126 MHz, CDCl_3_): *δ*(ppm) = 161.13 and 159.23 (C_1_-F, *J*_C-F_ = 239.14 Hz), 136.22 and 136.12 (C_5_-F, *J*_C-F_ = 10.34 Hz), 134.11, 129.61, 123.04, 122.94 and 122.91 (C_4_-F, *J*_C-F_ = 3.65 Hz), 119.87 and 119.79 (C_3_-F, *J*_C-F_ = 10.21 Hz), 111.85, 109.05 and 108.86 (C_6_-F, *J*_C-F_ = 24.57 Hz), 97.85 and 97.64 (C_2_-F, *J*_C-F_ = 25.96 Hz), 76.39, 37.92; LC/MS (ESI): found 353.10 [M+H]^+^, C_16_H_11_Cl_2_FN_2_O_2_ requires 352.01; anal. calcd. for C_16_H_11_Cl_2_FN_2_O_2_: C, 54.41; H, 3.14; N, 7.93; found: C, 54.58; H, 3.08; N, 8.03.

#### 3.7.19. (*S*)-1-Ethyl-3-(1-(4-fluorophenyl)-2-nitroethyl)-1*H*-indole (**10s**)

1-Ethyl-1H-indole **8d** (29 mg, 0.2 mmol) and 4-floronitrostyrene **9a** (34 mg, 0.2 mmol) were reacted according to the **GP4** to yield product **10s** as yellow oil (isolated yield 46 mg, 73%). Enantiomeric excess (*ee*) was determined by chiral HPLC (Chiracel OD-H column) (70% *n*-hexane/*i*-PrOH, 1.0 mL/min; *t*_minor_ = 16.38 min; *t*_major_ = 34.92 min; λ = 254 nm); 35.2% *ee*; ${\left[\alpha \right]}_{D}^{20}=+44.27^{o}$ (*c* 0.022, CH_3_OH); IR (KBr): 3418, 1557, 1349, 1174, 739, 573 cm^-1^; ^1^H-NMR (500 MHz, CDCl_3_): *δ*(ppm) = 7.42 (d, *J* = 7.9 Hz, 1H, Ar–**H**), 7.36–7.29 (m, 3H, Ar–**H**), 7.25–7.21 (m, 1H, Ar–**H**), 7.08 (ddd, *J* = 8.0, 7.0, 1.0 Hz, 1H, Ar–**H**), 7.02 (t, *J* = 8.6 Hz, 2H, Ar–**H**), 6.92 (d, *J* = 0.9 Hz, 1H, Ar–**H**), 5.18 (dd, *J* = 8.7, 7.4 Hz, 1H, C**H**), 5.06 (dd, *J* = 12.5, 7.2 Hz, 1H, C**H**_2(a)_), 4.91 (dd, *J* = 12.5, 8.9 Hz, 1H, C**H**_2(b)_), 4.14 (q, *J* = 7.3 Hz, 2H, C**H**_2_), 1.45 (t, *J* = 7.3 Hz, 3H, C**H**_3_); ^13^C-NMR (126 MHz, CDCl_3_): *δ*(ppm) = 163.17 and 161.21 (C_1_-F, *J*_C-F_ = 246.71 Hz), 136.50, 135.28 and 135.26 (C_4_-F, *J*_C-F_ = 3.15 Hz), 129.52, 129.46 (C_3_-F, *J*_C-F_ = 8.06 Hz), 126.66, 124.56, 122.34, 119.62, 119.14, 116.02, 115.85 (C_2_-F, *J*_C-F_ = 21.55 Hz), 112.81, 109.79, 79.73, 41.19, 41.06, 15.55; LC/MS (ESI): found 313.10 [M+H]^+^, C_18_H_17_FN_2_O_2_ requires 312.13; anal. calcd. for C_18_H_17_FN_2_O_2_: C, 69.22; H, 5.49; N, 8.97; found: C, 69.34; H, 5.43; N, 8.85.

#### 3.7.20. (*S*)-1-Ethyl-3-(1-(4-methoxyphenyl)-2-nitroethyl)-1*H*-indole (**10t**)

1-Ethyl-1H-indole **8d** (29 mg, 0.2 mmol) and 4-methoxynitrostyrene **9d** (36 mg, 0.2 mmol) were reacted according to the **GP4** to yield product **10t** as yellow oil (isolated yield 49 mg, 76%). Enantiomeric excess (*ee*) was determined by chiral HPLC (Chiracel OD-H column) (70% *n*-hexane/*i*-PrOH, 1.0 mL/min; *t*_minor_ = 20.94 min; *t*_major_ = 35.44 min; λ = 254 nm); 26.74% *ee*; ${\left[\alpha \right]}_{D}^{20}=+20.60^{o}$ (*c* 0.024, CH_3_OH); IR (KBr): 3417, 152, 1337, 1171, 741, 534 cm^−1^; ^1^H-NMR (500 MHz, CDCl_3_): *δ*(ppm) = 7.44 (d, *J* = 8.0 Hz, 1H, Ar–**H**), 7.32 (d, *J* = 8.3 Hz, 1H, Ar–**H**), 7.27–7.24 (m, 2H, Ar–**H**), 7.21 (t, *J* = 7.0 Hz, 1H, Ar–**H**), 7.06 (t, *J* = 7.5 Hz, 1H, Ar–**H**), 6.90 (s, 1H, Ar–**H**), 6.85 (d, *J* = 8.7 Hz, 2H, Ar–**H**), 5.13 (t, *J* = 8.0 Hz, 1H, C**H**), 5.03 (dd, *J* = 12.3, 7.3 Hz, 1H, C**H**_2(a)_), 4.89 (dd, *J* = 12.4, 8.8 Hz, 1H, C**H**_2(b)_), 4.12 (q, *J* = 7.3 Hz, 2H, C**H**_2_), 3.77 (s, 3H, C**H**_3_), 1.43 (t, *J* = 7.3 Hz, 3H, C**H**_3_); ^13^C-NMR (126 MHz, CDCl_3_): *δ*(ppm) = 158.99, 136.49, 131.51, 128.94, 126.82, 124.62, 122.18, 119.48, 119.28, 114.38, 113.33, 109.70, 55.37, 41.14, 41.06, 15.55; LC/MS (ESI): found 325.20 [M+H]^+^, C_19_H_20_N_2_O_3_ requires 324.15; anal. calcd. for C_19_H_20_N_2_O_3_: C, 70.35; H, 6.21; N, 8.64; found: C, 70.19; H, 6.13; N, 8.54.

## 4. Large-Scale Synthesis of (*S*)-3-(1-(4-Fluorophenyl)-2-nitroethyl)-1*H*-indole (10a)

An oven-dried 50-mL round bottom flask equipped with a condenser under nitrogen atmosphere was charged with ligand **L5** (100 mg, 0.3 mmol, 15% mol), Cu(OTf)_2_ (110 mg, 0.3 mmol, 15 mol %) and dry toluene (20 mL). The mixture was then stirred at reflux for 2 h. After cooling to room temperature, 4-floronitrostyrene (**9a)** (334 mg, 2.0 mmol) and 4A° molecular sieves were added. Then, the mixture was stirred for another 30 min, followed by the addition of indole **8a** (234 mg, 2.0 mmol). The reaction was then left stirring for 48 h at room temperature. The solvent was removed under reduced pressure, and the crude product was isolated by flash column chromatography on silica gel, eluting with ethylacetate/*n*-hexane (2:8, *v*/*v*) to afford a pure Friedel–Crafts product (**10a**) isolated yield of 76% (432 mg) with 77.2% enantiomeric excess (*ee*). Enantiomeric excess (*ee*) was determined by chiral HPLC (Chiracel OD-H column) (70% *n*-hexane/*i*-PrOH, 1.0 mL/min; *t*_major_ = 25.09 min; *t*_minor_ = 30.30 min; λ = 254 nm); 77.2% *ee*; ^1^H-NMR (500 MHz, CDCl_3_): *δ*(ppm) = 8.13 (s, 1H, N**H**), 7.47–7.40 (m, 1H, Ar–**H**), 7.35 (s, 1H, Ar–**H**), 7.32–7.28 (m, 2H, Ar–**H**), 7.23 (ddd, *J* = 8.2, 7.0, 1.2 Hz, 1H, Ar–**H**), 7.11 (ddd, *J* = 8.1, 6.9, 1.0 Hz, 1H, Ar–**H**), 7.04–6.96 (m, 3H, Ar–**H**), 5.19 (t, *J* = 8.0 Hz, 1H, C**H**), 5.05 (dd, *J* = 12.5, 7.5 Hz, 1H, C**H**_2(a)_), 4.90 (dd, *J* = 12.5, 8.6 Hz, 1H, C**H**_2(b)_); ^13^C-NMR (126 MHz, CDCl_3_): *δ*(ppm) = 163.1 and 161.18 (C_1_-F, *J*_C-F_ = 246.58 Hz), 136.6, 135.07 and 135.04 (C_4_-F, *J*_C-F_ = 3.15 Hz), 129.50 and 129.44 (C_3_-F, *J*_C-F_ = 7.94 Hz), 126.0, 122.9, 121.6, 120.1, 118.9, 115.99 and 115.82 (C_2_-F, *J*_C-F_ = 21.67 Hz), 114.2, 111.6, 79.6, 41.0.

## 5. Conclusions

In summary, we have synthesized new *C*_2_-symmetric 2,5-*bis*(oxazolinyl)thiophene and 2,5-*bis*(imidazolinyl)thiophene ligands based on thiophene systems and successfully tested them in asymmetric Friedel–Crafts alkylation reactions of indole with trans *β*-nitroolefins. Our newly developed catalytic system (15 mol% of **L5**:Cu(OTf)_2_ in toluene at 25 °C) was found to be applicable in inducing chirality into nitroalkylated indoles with low to good yields (35–76%) and low to good enantioselectivity (21–81%) at room temperature. On the basis of the screening performed, this methodology could be an alternative tool for asymmetric Friedel–Crafts reactions using this catalytic system. The advantage of this catalytic system is that it is easy to prepare the chiral ligands from the widely accessible thiophene precursor, and the reaction can also be performed at room temperature as compared to other catalytic system carried out at lower temperatures. There is an ongoing research project to explore more utilities for these new chiral thiophene ligands and their applications in asymmetric transformation, and its outcome will be communicated soon in future.

## Data Availability

The data presented in this study are available in Appendix A.

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
