# Peer review of "Exploiting the Chiral Ligands of Bis(imidazolinyl)- and Bis(oxazolinyl)thiophenes—Synthesis and Application in Cu-Catalyzed Friedel–Crafts Asymmetric Alkylation"

_molecules, 2021, doi:10.3390/molecules26237408_

Round 1

Reviewer 1 Report

The paper describes the synthesis and evaluation of novel bis(imidazolinyl)- and bis(oxazolinyl)thiophenes as chiral ligands in Cu-catalyzed Friedel-Crafts asymmetric alkylation. By using well known and classical reactions, the bis-imidazoline-based ligands were synthesized in quite lower yields (35-40%) after five steps, starting from thiophene-2,5-dicarboxylic acid. When combining these ligands with Cu(OTf)2 in toluene (the solvent of choice for the alkylation reactions) at room temperature in F-C alkylations, good yields could be reached for the expected products. However, they were isolated in almost racemic forms. In other hands, the bis-oxazoline-based ligands (prepared with improved yields in three steps from the same starting material), displayed better results in terms of enantiomeric induction. The alkylations seem very sensitive to both solvent and metal sources. The scope is a bit narrow (12 products; 35-67% yields; 63-81% ee) and covers the combination of only NH-free indoles with aromatic nitrostyrene derivatives. It would be appreciated to have some outcomes on trials run with other substrates (such as N-protected indoles; C2-substituted indoles; aliphatic nitrostyrene derivatives…). Some post-functionalization processes and a large-scale synthesis procedure would also be interesting to broaden the applicability of the method. The paper needs English polishing; some words are incorrect and some sentences are confusing.

For this referee, the manuscript might be acceptable for a future publication in Molecules-MDPI, but only after the points raised above are adequately taken into consideration (scope; post-functionalization; scale-up; English polishing). In addition, some other imprecise points also deserve clarifications/corrections:

- The authors have mentioned that the chiral ligands were prepared with “excellent optical purity” or that they are “optically pure ligands”. How can they prove it? Even though they have used enriched amino alcohols 3 during the synthesis of the ligands, they have to provide chiral HPLC analysis for all new ligands in order to prove that they did not suffer from some enantiomeric erosion during their preparation.

- If compounds 10g, 10h, 10i, 10k and 10l are new ones, please provide IR and HRMS information; otherwise cite the references where we can find it.

- The arrows on the proposed mechanistic pathway (catalytic cycle) are not the good ones.

- Please, provide a better chiral HPLC profile for compound 10h.

- It would be appreciable to have a general procedure described for the synthesis of racemic F-C alkylation products (even though the authors have used reported methods). 

- On the conclusion, the authors have written “On the basis of the screening performed, this methodology can be provided a generalized summary for which Friedel-Crafts reactions using this catalytic system is more suitable.” I am not convinced that it is completely true; I believe that the method might be an alternative to the ones already described. Indeed, in some cases (mainly by using organocatalysis), the ees are higher and the scope is broader than the ones obtained with the conditions described in the submitted manuscript. Please, either tone down the arguments on this sentence or prove that other strategies (by adding a scheme for example) are less efficient than theirs.

Author Response

Response to Reviewer 1_round1 Comments

Point 1: Comments and Suggestions for Authors:

The paper describes the synthesis and evaluation of novel bis(imidazolinyl)- and bis(oxazolinyl)thiophenes as chiral ligands in Cu-catalyzed Friedel-Crafts asymmetric alkylation. By using well known and classical reactions, the bis-imidazoline-based ligands were synthesized in quite lower yields (35-40%) after five steps, starting from thiophene-2,5-dicarboxylic acid. When combining these ligands with Cu(OTf)2 in toluene (the solvent of choice for the alkylation reactions) at room temperature in F-C alkylations, good yields could be reached for the expected products. However, they were isolated in almost racemic forms. In other hands, the bis-oxazoline-based ligands (prepared with improved yields in three steps from the same starting material), displayed better results in terms of enantiomeric induction. The alkylations seem very sensitive to both solvent and metal sources. The scope is a bit narrow (12 products; 35-67% yields; 63-81% ee) and covers the combination of only NH-free indoles with aromatic nitrostyrene derivatives. It would be appreciated to have some outcomes on trials run with other substrates (such as N-protected indoles; C2-substituted indoles; aliphatic nitrostyrene derivatives…). Some post-functionalization processes and a large-scale synthesis procedure would also be interesting to broaden the applicability of the method. The paper needs English polishing; some words are incorrect and some sentences are confusing.

For this referee, the manuscript might be acceptable for a future publication in Molecules-MDPI, but only after the points raised above are adequately taken into consideration (scope; post-functionalization; scale-up; English polishing). In addition, some other imprecise points also deserve clarifications/corrections:

Response 1. We have included 8 example more in the substrate scope (total 20 examples). Taking consideration of your advice we have used N-ethyl protected indole, 2-thienylnitrostyrene, more hindered 2,6-Cl2 phenyl-nitrostyrene and 10 fold scale up the reaction in one example etc and the results are listed in the table 3. Moreover we have revised the manuscript thoroughly.

Point 2: - The authors have mentioned that the chiral ligands were prepared with “excellent optical purity” or that they are “optically pure ligands”. How can they prove it? Even though they have used enriched amino alcohols 3 during the synthesis of the ligands, they have to provide chiral HPLC analysis for all new ligands in order to prove that they did not suffer from some enantiomeric erosion during their preparation.

Response 2: All the ligands optical rotation have been measured then we checked the HPLC purity for all the ligands. It is found that all the ligands are more than 99% pure.

Point 3: - If compounds 10g, 10h, 10i, 10k and 10l are new ones, please provide IR and HRMS information; otherwise cite the references where we can find it.

Response 3: IR and HRMS data have been included for all new compounds.

Point 4: - The arrows on the proposed mechanistic pathway (catalytic cycle) are not the good ones.

Response 4: Has been corrected.

Point 5: - Please, provide a better chiral HPLC profile for compound 10h.

Response 5: Better chiral HPLC profile for compound 10h has been provided.

Point 6: - It would be appreciable to have a general procedure described for the synthesis of racemic F-C alkylation products (even though the authors have used reported methods). 

Response 6: General procedure described for the synthesis of racemic F-C alkylation products has been included.

Point 7: - On the conclusion, the authors have written “On the basis of the screening performed, this methodology can be provided a generalized summary for which Friedel-Crafts reactions using this catalytic system is more suitable.” I am not convinced that it is completely true; I believe that the method might be an alternative to the ones already described. Indeed, in some cases (mainly by using organocatalysis), the ees are higher and the scope is broader than the ones obtained with the conditions described in the submitted manuscript. Please, either tone down the arguments on this sentence or prove that other strategies (by adding a scheme for example) are less efficient than theirs.

Response 7: You are absolutely correct and we have modified accordingly.

Reviewer 2 Report

In the manuscript (molecules-1468716), Islam et al. reported the synthesis and characterization of a set of five symmetric chiral ligands of 2,5 bis-(imidazolinyl) /(oxazolinyl) thiophene. The authors explored the applicability of one of 2,5-bis(oxazolinyl)thiophene ligands to promote and catalyze the asymmetric alkylation via Cu-catalyzed Friedel-Craft reaction. The authors showed that under optimized reaction conditions, the alkylated products were obtained in good yields and high enantioselectivity. Finally, the authors  proposed a mechanism of action for the ligand in catalyzing the Friedel Craft alkylation reaction. Overall, this is a very interesting study which is well-structured and well performed.   However, there are some concerns that the authors should address:

1- C-NMR for L1-3, the resolution is very low and many peaks are not visible. Please submit high-resolution spectra. Also, where the spectra for 5a, 7a-c?

2- why the authors have not measured the optical activity for the synthesized ligands?

3- It would be suggested that the authors include in the introduction part some structures of previously potent catalysts of this class of compounds

4- Indeed, the design of the synthesized ligands is not clear. WHy the authors decided i-Bu and sec-Bu for the oxazoliny-moiety and i-pr for imidazolinyl-moiety?

5- it is clear from the results that the oxazoliny-based ligands are more potent than imidazolinyl-based ones, more interestingly the substitution at oxazoliny-moiety showed to be critical, the authors should discuss this in more details and discuss how this class of ligand could be modified in future for better activity.

6- I think the main drawback of this study that the authors have ONLY selected different substituent of the same substrate (nitrostyrene). It's clear that the nitrostyrene is a very active substrate (group) toward Friedel Craft alkylation. One would argue about the reactivity of the present ligand (L5) for other substrates, e.g, styrene?? Why the authors used just one scaffold?? the same for the indol-part, it is very active substrate for FCA. Therefore, we can NOT conclude the real applicability/generality of the reported ligand (L5).

7- To the same line, this is also supported by the fail occurred in the synthesis of 10m and 10n compounds.

8- In line 546, the authors claimed the synthesis and characterization of two nitostyrene based indole scaffold 10m and 10n? However, it was not successful, accordingly please modify this paragraph to be more clear (546-548).

9- Indeed, I'm not convinced with the authors' argue about the X-ray of compounds 9h and 9g as a reason for the unreactivity. Indeed, lines 558-568 have no clear argue which is related to reaction mechanism.

Author Response

Response to Reviewer 2_round1 Comments

Comments and Suggestions for Authors

In the manuscript (molecules-1468716), Islam et al. reported the synthesis and characterization of a set of five symmetric chiral ligands of 2,5 bis-(imidazolinyl) /(oxazolinyl) thiophene. The authors explored the applicability of one of 2,5-bis(oxazolinyl)thiophene ligands to promote and catalyze the asymmetric alkylation via Cu-catalyzed Friedel-Craft reaction. The authors showed that under optimized reaction conditions, the alkylated products were obtained in good yields and high enantioselectivity. Finally, the authors  proposed a mechanism of action for the ligand in catalyzing the Friedel Craft alkylation reaction. Overall, this is a very interesting study which is well-structured and well performed.   However, there are some concerns that the authors should address:

Point 1: C-NMR for L1-3, the resolution is very low and many peaks are not visible. Please submit high-resolution spectra. Also, where the spectra for 5a, 7a-c?

Response 1: C-NMR for L1-3 has been improved.

Since, compound 5a, 7a-c were not isolated and directly taken to next step so NMR not available.

Point 2: why the authors have not measured the optical activity for the synthesized ligands?

Response 2: Optical rotation has been measured and already written in the manuscript.

Point 3: It would be suggested that the authors include in the introduction part some structures of previously potent catalysts of this class of compounds.

Response 3: Previously reported few potent ligands structure has been included in the introduction section in figure 1.

Point 4: Indeed, the design of the synthesized ligands is not clear. WHy the authors decided i-Bu and sec-Bu for the oxazoliny-moiety and i-pr for imidazolinyl-moiety?

Response 4: Since the research work is part of PhD students, we have designed these ligands initially for the synthesis of some chiral ligands based on thiophene moiety and their chiral applications. But we still exploring another ligands and also your suggestion well be considered highly appreciated your idea.

Point 5: it is clear from the results that the oxazoliny-based ligands are more potent than imidazolinyl-based ones, more interestingly the substitution at oxazoliny-moiety showed to be critical, the authors should discuss this in more details and discuss how this class of ligand could be modified in future for better activity.

Response 5: Thanks for your observation; we have added this statement in the manuscript. We still working in this area for better understanding why this substitution at oxazoliny-moiety play a crucial role for asymmetric induction.

“Interestingly, it is clear from the preliminary results that the oxazoliny-based ligands are more potent than imidazolinyl-based ones, more interestingly the substitution at oxazoliny-moiety showed to be also critical for the asymmetric induction”.

Point 6: I think the main drawback of this study that the authors have ONLY selected different substituent of the same substrate (nitrostyrene). It's clear that the nitrostyrene is a very active substrate (group) toward Friedel Craft alkylation. One would argue about the reactivity of the present ligand (L5) for other substrates, e.g, styrene?? Why the authors used just one scaffold?? the same for the indol-part, it is very active substrate for FCA. Therefore, we can NOT conclude the real applicability/generality of the reported ligand (L5).

Response 6: We have included 8 more example of FC reaction with 2-thienyl-nitrostyrene, more hindered 2,6-dichloronitrostyre and 5-fluoro indole as well N-ethyl protected indole and the findings are summarised in table 3. We tried the reaction with nitrostyrene based on indole moiety but unfortunately reaction did not succeeded.

Point 7: To the same line, this is also supported by the fail occurred in the synthesis of 10u and 10v compounds.

Response 7: Accordingly we have modified the sentence in conclusion part.

Point 8: In line 718, the authors claimed the synthesis and characterization of two nitostyrene based indole scaffold 10u and 10v? However, it was not successful, accordingly please modify this paragraph to be more clear (718-723).

Response 8: revised this paragraph accordingly.

Point 9: Indeed, I'm not convinced with the authors' argue about the X-ray of compounds 9i and 9j as a reason for the unreactivity. Indeed, lines 722-728 have no clear argue which is related to reaction mechanism.

Response 9: Thanks for your comment, just to clarify luckily we got the crystalline compounds for the nitro-styrene so we have added this as confirmation or the structure only not a reason for the un-reactivity. Ok, now for misunderstanding we moved the crystal data into SI.

Reviewer 3 Report

This manuscript reveals the details of research into the use of substituted bis(imidazolinyl)- and bis(oxazolinyl)-thiophenes as ligands in copper bis-tosylate complexes catalyzing Friedel-Crafts Asymmetric Alkylation of indole with substituted nitrostyrenes.

In general, the investigation itself is well-performed, a big experimental work was done. But, the most important question appears after reading this manuscript: which new achievements were made in this work? The catalytic system for Friedel-Crafts Asymmetric Alkylation which includes Cu(OTf)2,  bis(imidazolinyl)- and bis(oxazolinyl)-thiophenes, -pyridines is well-known and well-documented. There are no new products of the nitrostyrenes addition to indoles (and authors say about this, e.g. referencing the literature sources in the experimental part: refs. 40, 41, 74-77). The catalytic activity and the enantiomeric excess found on these new ligands are not better than the described earlier for Friedel-Crafts Asymmetric Alkylation by the chiral aziridine-phosphines (ref. Catalysts 2020, 10, 971; doi:10.3390/catal10090971), by bis(oxazoline)pyridine-copper(II) complexes (ref. Tetrahedron Letters, 2007, 48, 1127; doi:10.1016/j.tetlet.2006.12.081), or, expectedly, by the identical and very close bis(oxazolinyl)thiophene-copper(II) complexes (ref. Catalysis Letters, 2014, 144, 943; DOI 10.1007/s10562-014-1228-2). Thus, to make the results much more important, as a minimum, more other substrates should be tested leading to new alkylation products.

 So, from the reviewer's point of view, there is not enough novelty and importance in the present manuscript to merit the publication in Molecules.

However, this is a well-performed work, and I suggest authors transfer this manuscript to one of the other related mdpi journals, for example, Processes, Reactions, or Symmetry.  

Some minor moments:

1. in the present state it is completely unclear from the abstract which substrates are involved in the  Friedel-Crafts Asymmetric Alkylation. The revision of the abstract is strongly recommended. 

2. Figure 3 about the proposed mechanism:  in the intermediate III, the direction of H+ transfer is wrong. It should be vice-versa.

3. An excessive numbering (even numbering of reagents and starting compounds) leads to clutter and confusion which complicates the reader's perception of the material.

Author Response

Response to Reviewer 3_round1 Comments

Comments and Suggestions for Authors

This manuscript reveals the details of research into the use of substituted bis(imidazolinyl)- and bis(oxazolinyl)-thiophenes as ligands in copper bis-tosylate complexes catalyzing Friedel-Crafts Asymmetric Alkylation of indole with substituted nitrostyrenes.

In general, the investigation itself is well-performed, a big experimental work was done. But, the most important question appears after reading this manuscript: which new achievements were made in this work? The catalytic system for Friedel-Crafts Asymmetric Alkylation which includes Cu(OTf)2, bis(imidazolinyl)- and bis(oxazolinyl)-thiophenes, -pyridines is well-known and well-documented. There are no new products of the nitrostyrenes addition to indoles (and authors say about this, e.g. referencing the literature sources in the experimental part: refs. 40, 41, 74-77). The catalytic activity and the enantiomeric excess found on these new ligands are not better than the described earlier for Friedel-Crafts Asymmetric Alkylation by the chiral aziridine-phosphines (ref. Catalysts 2020, 10, 971; doi:10.3390/catal10090971), by bis(oxazoline)pyridine-copper(II) complexes (ref. Tetrahedron Letters, 2007, 48, 1127; doi:10.1016/j.tetlet.2006.12.081), or, expectedly, by the identical and very close bis(oxazolinyl)thiophene-copper(II) complexes (ref. Catalysis Letters, 2014, 144, 943; DOI 10.1007/s10562-014-1228-2). Thus, to make the results much more important, as a minimum, more other substrates should be tested leading to new alkylation products.

 So, from the reviewer's point of view, there is not enough novelty and importance in the present manuscript to merit the publication in Molecules.

However, this is a well-performed work, and I suggest authors transfer this manuscript to one of the other related mdpi journals, for example, Processes, Reactions, or Symmetry.  

Some minor moments:

Point 1: in the present state it is completely unclear from the abstract which substrates are involved in the Friedel-Crafts Asymmetric Alkylation. The revision of the abstract is strongly recommended. 

Response 1: The abstract has been revised.

Point 2: Figure 3 about the proposed mechanism:  in the intermediate III, the direction of H+ transfer is wrong. It should be vice-versa.

Response 2: The direction of H+ transfer has been corrected.

Point 3: An excessive numbering (even numbering of reagents and starting compounds) leads to clutter and confusion which complicates the reader's perception of the material.

Response 3: we apologize that the numbering is confusing. We have done our best for better clarity.

Round 2

Reviewer 1 Report

The revised version of the ms "Exploiting the Chiral Ligands of Bis(imidazolinyl)- and Bis(oxazolinyl)thiophenes - Synthesis and Application in Cu-Catalyzed Friedel-Crafts Asymmetric Alkylation" should be now suitable for publication into Molecules. However, a few points deserves corrections/improvements:

  • racemic HPLC profiles for ligands L1-L5 have to be given (not only the chiral ones),
  • even though the authors have revised the manuscript thoroughly (English polishing), some sentences are still not very clear (for example:  the two sentences highligted with yellow background on the abstract).

Only after taking into consideration the above mentioned points, the manuscript can be accepted for publication.

Reviewer 2 Report

Thanks to the authors for addressing all concerns that have been raised. The manuscript has been significantly modified. I would recommend the publication of this study.

Reviewer 3 Report

Authors have made the corrections in the accordance with the comments of the first review round, and have included some new reactants to the investigations of the catalytic activity of copper complex. In light of the performed improvements in the manuscript, I tend to change my opinion, the article may be accepted for the publication. However, please pay attention to English before the final publication (e.g. the first line in the abstract: " Five of a new symmetric chiral ligands..." etc..).

Good luck in the future investigations!